



# A modern snapshot of the isotopic composition of lacustrine biogenic carbonates – Records of seasonal water temperature variability

Inga Labuhn[1,*], Franziska Tell[1,2,*], Ulrich von Grafenstein[3], Dan Hammarlund[4], Henning Kuhnert[2], and Bénédicte Minster[3]

[1]Institute of Geography, University of Bremen, Celsiusstr. 2, 28359 Bremen, Germany
[2]MARUM – Center for Marine Environmental Sciences, University of Bremen, Leobener Str. 8, 28359 Bremen, Germany
[3]Laboratoire des Sciences du Climat et de l'Environnement, LSCE/IPSL, CEA-CNRS-UVSQ, Université Paris-Saclay, Gif-sur-Yvette, France
[4]Quaternary Sciences, Department of Geology, Lund University, Sölvegatan 12, SE-223 62 Lund, Sweden
[*]These authors contributed equally to this work.

**Correspondence:** Inga Labuhn (labuhn@uni-bremen.de)

**Abstract.**

Carbonate shells and encrustations from lacustrine organisms provide proxy records of past environmental and climatic changes. The carbon isotopic composition ($\delta^{13}C$) of such carbonates depends on the $\delta^{13}C$ of dissolved inorganic carbon (DIC). Their oxygen isotopic composition ($\delta^{18}O$) is controlled by the $\delta^{18}O$ of the lake water and on water temperature during

carbonate precipitation. Lake water $\delta^{18}O$, in turn, reflects the $\delta^{18}O$ of precipitation in the catchment, water residence time and mixing, and evaporation. A paleoclimate interpretation of carbonate isotope records requires a site-specific calibration based on an understanding of these local conditions.

For this study, samples of different carbonate components and water were collected in the littoral zone of Lake Locknesjön, central Sweden (62.99°N, 14.85°E, 328 m a.s.l.) along a water depth gradient from 1 to 8 m. Samples from living organisms

and sub-recent samples in surface sediments were taken from the calcifying alga *Chara hispida*, mollusks from the genus *Pisidium*, and adult and juvenile instars of two ostracod species, *Candona candida* and *Candona neglecta*.

Neither the isotopic composition of carbonates nor the $\delta^{18}O$ of water vary significantly with water depth, indicating a well-mixed epilimnion. The mean $\delta^{13}C$ of *Chara hispida* encrustations is 4‰ higher than the other carbonates. This is due to fractionation related to photosynthesis, which preferentially incorporates $^{12}C$ in the organic matter and increases the $\delta^{13}C$ of the

encrustations. A small effect of photosynthetic $^{13}C$ enrichment in DIC is seen in contemporaneously formed valves of juvenile ostracods. The largest differences in the mean carbonate $\delta^{18}O$ between species are caused by vital offsets, i.e. the species-specific deviations from the $\delta^{18}O$ of inorganic carbonate which would have been precipitated in isotopic equilibrium with the water. After subtraction of these offsets, the remaining differences in the mean carbonate $\delta^{18}O$ between species can mainly be attributed to seasonal water temperature changes. The lowest $\delta^{18}O$ values are observed in *Chara hispida* encrustations,

which form during the summer months when photosynthesis is most intense. Adult ostracods, which calcify their valves during the cold season, display the highest $\delta^{18}O$ values. This is because an increase in water temperature leads to a decrease in fractionation between carbonate and water, and therefore to a decrease in carbonate $\delta^{18}O$. At the same time, an increase in air





temperature leads to an increase in the $\delta^{18}$O of lake water through its effect on precipitation $\delta^{18}$O and on evaporation from
the lake, and consequently to an increase in carbonate $\delta^{18}$O, opposite to the effect of increasing water temperature on oxygen-
isotope fractionation. However, the seasonal and inter-annual variability in lake water $\delta^{18}$O is small ($\sim$0.5‰) due to the long
water residence time of the lake. Seasonal changes in the temperature-dependent fractionation are therefore the dominant cause
of carbonate $\delta^{18}$O differences between species when vital offsets are corrected.

Temperature reconstructions based on paleotemperature equations for equilibrium carbonate precipitation using the mean
$\delta^{18}$O of each species and the mean $\delta^{18}$O of lake water are well in agreement with the observed seasonal water temperature
range. The high carbonate $\delta^{18}$O variability of samples within a species, on the other hand, leads to a large scatter in the
reconstructed temperatures based on individual samples. This implies that care must be taken to obtain a representative sample
size for paleotemperature reconstructions.

## 1  Introduction

The oxygen and carbon isotopic compositions of biogenic carbonates from lake sediments are powerful tools for reconstruc-
tions of past climatic and environmental changes. Despite the complex influences of climate, hydrology, catchment and lake
processes, as well as biological effects, they may be interpreted – depending of the local context – as proxies of temperature,
evaporation, the isotopic composition of precipitation, or atmospheric circulation patterns (e.g. Holmes et al., 2020; Andersson
et al., 2010; Jonsson et al., 2010; Hammarlund et al., 2003; Schwalb, 2003; Hammarlund et al., 2002; von Grafenstein et al.,
1999a).

Calcium carbonate ($CaCO_3$) remains from lacustrine organisms can be identified based on the characteristic morphology of
shells, valves or encrustations. For paleoclimate reconstructions, it is important to assess the isotopic composition of specific
biogenic carbonates to avoid the difficulties involved in the interpretation of isotopic records obtained on bulk carbonate matter
in lake sediments. Changes in the bulk carbonate composition can be caused by variations in the assemblage of calcifying
organisms, in clastic carbonate input, or in the amount of inorganic calcite precipitated in the lake (e.g. Bright et al., 2006;
Hammarlund and Buchardt, 1996).

The $\delta^{18}$O and $\delta^{13}$C of lacustrine biogenic carbonates depend on the isotopic composition of the oxygen and carbon sources,
water and dissolved inorganic carbon (DIC), respectively. The isotopic composition of these sources can be variable over space
and time, and is modified during carbonate precipitation, mainly through temperature-dependent fractionation and fractionation
during physiological processes.

The $\delta^{18}$O of biogenic carbonates carries the isotopic signature of the lake water, which, in turn, depends on the $\delta^{18}$O of
precipitation in the catchment, and is increased by evaporation from the lake (Gat, 1996). Elevated warm-season temperatures
lead to a higher $\delta^{18}$O of lake water because both the $\delta^{18}$O of precipitation and the evaporative enrichment of lake water increase
with temperature. Biogenic carbonates forming at different water depths may differ in their $\delta^{18}$O, e.g. only those formed above
the thermocline are affected by higher temperatures and evaporation during times of thermal stratification of the lake.





The $\delta^{13}$C of biogenic carbonates carries the isotopic signature of DIC (McConnaughey, 1989; Ito, 2001). It depends on the $\delta^{13}$C signal of DIC in recharging groundwater that is transferred into the lake. The $\delta^{13}$C of groundwater DIC is controlled by exchanges between atmospheric $CO_2$, vegetation/soil air $CO_2$, and carbonate bedrock under open or closed system conditions (Clark and Fritz, 1997). In lake water, the $\delta^{13}$C of DIC can vary seasonally, depending on photosynthetic activity in the photic zone, the oxidation of organic matter, and thermal stratification.

The fractionation of oxygen isotopes between water and carbonate is temperature dependent (Grossman and Ku, 1986; Kim and O'Neil, 1997; Beck et al., 2005; Kim et al., 2007). Biogenic carbonates formed during different times of the year can therefore differ in $\delta^{18}$O due to seasonal water temperature changes, which are especially strong in shallow water. Fractionation factors for calcite precipitated in isotopic equilibrium with the water are known from empirical studies. The $\delta^{18}$O of equilibrium carbonate decreases as water temperature increases, by -0.24‰°$C^{-1}$ (Craig and Gordon, 1965; Kim and O'Neil, 1997). Note that the relationship between precipitation $\delta^{18}$O and air temperature is opposite, with a $\delta^{18}$O increase of ∼0.6‰°$C^{-1}$ in mid latitudes (Rozanski et al., 1992). Kinetic effects on fractionation must be taken into account in the case of fast calcite precipitation, observed e.g. in *Chara* due to intense photosynthesis (Apolinarska et al., 2016; Andrews et al., 2004), which lowers the carbonate $\delta^{18}$O. Furthermore, variations in the pH have an influence on the isotopic composition of carbonates, with an increasing pH lowering the $\delta$-values (Herbst et al., 2018; Teranes et al., 1999; McConnaughey, 1989).

Fractionation due to physiological processes in a calcifying organism leads to a different isotopic composition of the biogenic calcite compared to inorganic calcite precipitated in isotopic equilibrium. This species-specific vital offset can be quantified through laboratory studies under controlled conditions (Li and Liu, 2010; Chivas et al., 2002; Xia et al., 1997b) or through monitoring in the field (Börner et al., 2017; Decrouy et al., 2011b; Keatings et al., 2002; von Grafenstein et al., 1999b). In addition to the vital offsets, the conditions in the micro-habitat of the carbonate-producing species can play a role as well (Decrouy et al., 2011a; Xia et al., 1997a). If the vital offsets for a given species is known and appears constant, it can be corrected for and does not hamper proxy interpretations. However, their causes are not yet fully understood, and it has been suggested that varying water chemistry has an influence on the vital effect (Devriendt et al., 2017; Decrouy et al., 2011b).

Recent studies have highlighted the seasonal bias in proxies of past sea surface temperatures and the importance of distinguishing between annual and seasonal temperature reconstructions (Bova et al., 2021; Hertzberg, 2021). Lacustrine biogenic carbonates formed during different times of the year can provide insights into seasonal temperature variations. In this context, seasonal influences under modern conditions must be understood and potential past water depth changes must be taken into account, because these could impact the location of carbonate formation with respect to a thermocline, changing the relationship between carbonates from organisms living at times of lake water stratification vs. at times of mixed water column.

For the use of $\delta^{18}$O and $\delta^{13}$C in lacustrine biogenic carbonates as paleoenvironmental proxies, these different influencing factors must be known or constrained. For example, benthic ostracods in deep lakes living below the permanent thermocline form their calcitic valves at a nearly constant temperature around 4 °C, close to the density maximum of freshwater. Variations in their $\delta^{18}$O therefore primarily reflect variations in the $\delta^{18}$O of precipitation if evaporative enrichment in $^{18}$O of the lake water is negligible (von Grafenstein and Labuhn, 2021). Under such optimal conditions, the $\delta^{18}$O of ostracod valves provides the basis for reconstructing $\delta^{18}$O in precipitation (e.g. von Grafenstein et al., 1999a). However, ostracod population densities





are usually low and carbonates are not always preserved in profundal sediments. Carbonate-producing organisms (including algae that need light for photosynthesis) are often more abundant at littoral sites, where the the interpretation of isotope proxies is complicated by a larger seasonal water temperature amplitude and a stronger influence of evaporation.

Disentangling the influences on the isotopic composition of specific biogenic carbonates is possible based on monitoring of the relevant parameters (such as temperature, $\delta^{18}O$ and pH of lake water, and the $\delta^{13}C$ of DIC), sufficiently long to represent

the entire time period of calcification of each organism and to capture seasonal and inter-annual variability. Monitoring can provide detailed understanding for interpretation of the proxies in the local context of each study (Börner et al., 2017; Meyer et al., 2017; Mayr et al., 2015), but long-term investigations (e.g. von Grafenstein et al., 1999b; Decrouy et al., 2011b) are rare and subject to logistic and financial constraints.

In this study, we therefore rely on a single sampling occasion to investigate the oxygen and carbon isotopic composition

($\delta^{18}O$ and $\delta^{13}C$) of different carbonate-producing species from Lake Locknesjön in central Sweden. At shallow water sites, the following types of samples were taken from living organisms and their sub-recent remains in surface sediments: the calcite encrustations of *Chara hispida*, the aragonitic shells of the mollusk *Pisidium sp.*, and the calcitic valves of two species of ostracods, *Candona candida* and *Candona neglecta*. In addition, two species of Charophyceae, *Chara aspera* and *Chara hispida* were sampled in the nearby Lake Blektjärnen.

Our goals are (1) to test whether this approach of a "snapshot" study with short-term sampling can narrow down the primary environmental drivers of the isotopic composition of specific biogenic carbonates when long-term monitoring studies are not feasible. We focus on inter- and intra-species variations and compare them against $\delta^{18}O$ of local precipitation, streams and lake water, considering vital offsets and the timing of calcification for each species. We further (2) assess the uncertainty in seasonal water temperature reconstructions based on isotope measurements of lake water and of carbonates from multiple species. Lastly

(3), this snapshot breaks down the details of what could be represented in a single measurement when building long isotope time series from sediment records. We discuss the sensitivity of such a measurement to the choice of the sampled material, and the implications for sub-sampling of sediment cores for paleoenvironmental studies. These investigations will also aid the site-specific paleoenvironmental interpretation of equivalent measurements from Holocene sediments in Lake Locknesjön.

## 2 Material and methods

### 2.1 Study site

Lake Locknesjön is located in central Sweden (62.99 °N, 14.85 °E, 328 m a.s.l., Figure 1). It is one of the few carbonate lakes in northern Scandinavia, and abundant biogenic carbonate deposits are found in its littoral sediments, with charophyte encrustations as the dominant sediment component. The lake has an area of 27 km$^2$, a maximum depth of 57.4 m and a mean depth of 17.6 m (SMHI, 2020a). The catchment area is 134 km$^2$, of which about 20% is the lake itself, 65% is coniferous

forest, 10% is agricultural land, and the rest is heathland, peat bogs and rural settlements (SMHI, 2020a). There are no other lakes upstream in the catchment.



The bedrock in the catchment area is dominated by Cambrian to Silurian carbonate-rich sedimentary rock, with some Precambrian crystalline bedrock in the southern part. The lake's central part lies within the Lockne Crater, which was formed by a meteorite impact 458 Ma ago (Ormö et al., 2010). The impact structure consists of impact breccia and resurge deposits (Sturkell et al., 2013). The fracture frequency of the breccia is estimated to be 25 times higher than in the surrounding crystalline bedrock, and the porosity is 2-10 times higher (Bäckström, 2005). These characteristics, together with the karstified carbonate bedrock, facilitate groundwater flow in the catchment (Dahlqvist et al., 2018). The Quaternary deposits consist mostly of glacial till with a typical thickness of 3-5 m (Dahlqvist et al., 2018). During the last deglaciation, the ice margin retreated north-westward from the study area around 10,000 cal. yr BP (Stroeven et al., 2015; Labuhn et al., 2018).

The hydrological model of the Swedish Meteorological and Hydrological Institute (SMHI) gives an average discharge at the outflow of Lake Locknesjön of 1.28 $\mathrm{m^3s^{-1}}$ (with a mean high water discharge of 3.10 $\mathrm{m^3s^{-1}}$, a mean low water discharge of 0.55 $\mathrm{m^3s^{-1}}$), and a residence time of 11.5 years. Inflow of surface water to the lake is limited to a few small streams. The modeled average discharge of the largest inflows combined is 0.37 $\mathrm{m^3s^{-1}}$ (SMHI, 2020a). Based on this modeled river discharge data, groundwater can be estimated to make up about 70% of the inflow to the lake. During our field work, groundwater inflows were identified based on a bubbling movement of fine sediment on the lake floor, which was observed on several locations along the sampled transect (see video supplement). Dahlqvist et al. (2018) describe traces of underwater erosion as further indications of groundwater inflow, with water temperatures at such inflows 6-8 °C lower than in the surrounding lake water.

Locknesjön is ice-covered from early December to mid-May (1961-1990 average; SMHI, 2020a). It is a dimictic lake, which is stratified during January and from July to September, and mixed between these periods (Figure 2a; Eklund, 1998). Autumn mixing occurred at temperatures of about 9 °C in 1957 (the only year for which monthly water temperature measurements are available). The average pH of the lake water is 8.16, with values up to 8.33 above the thermocline during summer (Figure 2b; data from the Triple Lakes Project, http://www.triplelakes.se/).

The average annual air temperature at the meteorological station of Mörsil, 70 km north-west of the lake, is 2.3 °C, and the average annual precipitation is 663 mm (1975-2005 averages; SMHI, 2020b). For the calcification of *Chara*, temperature and precipitation between May (ice melting) and July (sampling) are most relevant because this period corresponds to the time of photosynthetic activity. In the year of sampling (2018), the average May-July temperature was 2 °C higher and the May-July precipitation sum was 40 mm lower than in the reference period (Figure 3a). The modeled discharge at the lake's in- and outflow in 2018 shows no significant difference from the average conditions (SMHI, 2020a). The GNIP (Global Network of Isotopes in Precipitation) station closest to our study area, Bredkälen, is located 100 km north of Locknesjön at 400 m a.s.l. and provides monthly measurements of $\delta^{18}$O for the period of 1975-1980, but no measurements of $\delta$D (Figure 3b; IAEA/WMO, 2019).

## 2.2 Field sampling

In July 2018, samples of lake water, surface sediments and living *Chara hispida* were collected by divers at 13 locations along a transect from the western shoreline towards the center of Lake Locknesjön (Figure 1c). Water depths at the sampling locations





range from 1 to 8 m. Surface sediment samples of < 1 cm depth were taken with a small shovel and filled into sealable plastic bags. At 5.6 m water depth, water from a groundwater inflow was sampled with a syringe. Water samples were also collected at the outflow of Lake Locknesjön, and from surface water of the nearby Lake Blektjärnen (Figure 1b). Further water samples were taken in 2013 and 2014 at depths from 0 to 20 m at Lake Locknesjön (11 samples), at one of its main tributaries (Musån,
13 samples), and at its outflow (Forsaån, 6 samples). Living *Chara hispida* were only found down to 7 m water depth. For a comparison of different species of *Chara*, samples of *C. hispida* and *C. aspera* were collected at a depth of 0.3 m in Lake Blektjärnen. In Locknesjön only *C. hispida* were found.

## 2.3   Carbonate sample preparation

*Chara* species were determined based on the number of cells of the main stem and the differences between primary and sec-
ondary cells (Caisova and Gabka, 2009). A further indicator was the relation between the number of cells in the main stem and the number of branchlets in a whorl. Since the whorls of branchlets are not preserved in the sediments (only branchlet fragments), this was only useful for the determination of living algae. Internodes and branchlets were cut off from the living *Chara* and dried. The organic matter was removed from the calcite encrustations by putting the samples in 10% hydrogen peroxide ($H_2O_2$) for 48 h. They were then cleaned with a thin paintbrush to remove fine-grained calcite sticking to the encrustations.
The removed fine calcite was also kept for isotope measurements.

Sediment samples were wet sieved to obtain residual size fractions containing specific biogenic carbonates (>400 µm and >200 µm). Valves of ostracods and *Pisidium* and encrustations of *Chara* were picked out under a binocular microscope. Based on their size, the latter were assumed to be fragments of internodes. Ostracod species and *Pisidium* were identified based on the characteristics of their valves, following Meisch (2000) for ostracod identification. The dominant ostracod species
were *Candona candida* and *Candona neglecta*. Other species (*Cytherissa lacustris*, *Candona candata*, *Cypridopsis vidua*, and *Limnocythere inopinata*) were present in insufficient numbers for isotope measurements. Adult (A) and juvenile instars (A-1 to A-4) of ostracods were separated. The living specimens of ostracods and *Pisidium* found in the sediment samples were briefly boiled to detach valves easily. All valves were rinsed with ethanol and air dried (von Grafenstein et al., 1992; Mischke et al., 2007).

In order to obtain sufficient sample weights for isotope measurements, several ostracod valves from the same sampling location, species and instar were combined. Valves of living ostracods and subfossil specimens from the sediment samples were kept separate. The required numbers of valves in a sample were 2 for adults, 5 for A-1, 8 for A-2, and 15-25 for A-3 and A-4. If the sample weights were not sufficient due to few available valves, successive instars were combined for some locations, leading to combinations of A-2 and A-3, or A-3 and A-4 depending on the sample. *Pisidium* samples were composed
of individual valves. *Chara* samples were composed of single stalks from an internode or branchlet.

## 2.4   Isotope measurements

For this study, 49 water samples and 271 carbonate samples were measured. Measurements of $\delta^{18}O$ and $\delta D$ of water samples were conducted at the Laboratoire des Sciences du Climat et de l'Environnement (LSCE), Gif-sur-Yvette, France. The $\delta^{18}O$



was measured with a Finnigan MAT252 isotope ratio mass spectrometer with a precision of ±0.05‰ (two standard deviations).
The δD was measured with a Picarro Analyzer (Cavity Ring-Down Spectroscopy) with a precision of ±0.7‰ (one standard deviation). The results are given relative to the V-SMOW standard.

Carbonate $\delta^{18}$O and $\delta^{13}$C measurements were carried out at MARUM, Bremen, Germany, using a ThermoFisher Scientific 253plus gas isotope ratio mass spectrometer with a Kiel IV automated carbonate preparation device. The standard deviation of the reference standards over the measurement period was 0.06‰ for $\delta^{18}$O and 0.03‰ for $\delta^{13}$C. The results are given relative
to the V-PDB standard. Some sample weights were < 20 μg, resulting in low signal intensities (1800-3000 mV for mass 44). A control standard (ground limestone) showed no deviation over this signal range from the expected values within the above errors. 29 samples had a signal intensity <1800 mV and were excluded from further analyses. One major outlier in δ-values was identified in a sample of living *Candona neglecta*, instar A-2, taken at 5 m water depth (sampling location 5) and also excluded from further analysis. Its $\delta^{13}$C value of -9.63‰ was below the outer fence of the interquartile range of all ostracod
$\delta^{13}$C values.

## 2.5   Data analysis

Statistical analyses were performed using R v. 3.6.1 (R Core Team, 2018) and the statistical package PAST v. 3.25 (Hammer et al., 2001). All data sets of individual species were tested for normal distribution using the Shapiro-Wilk-Test (Shapiro and Wilk, 1965). For normally distributed data, correlations were calculated using the Pearson method (Pearson, 1895), and t-
tests assuming unequal variances were calculated to compare different sample sets. For samples lacking normal distribution, this comparison was done using the Mann-Whitney-Test (Mann and Whitney, 1947) and the Spearman method was used to calculate correlations (Spearman, 1904).

Assuming calcite precipitation in isotopic equilibrium with the water, theoretical water temperatures were calculated from the mean, minimum and maximum $\delta^{18}$O values of water and calcite (corrected for species-specific vital offsets). We used the
empirical equation by Kim and O'Neil (1997), with $\delta^{18}O_{calcite(SMOW)} - \delta^{18}O_{water(SMOW)}$ as an approximation of the fractionation factor, and the conversion of calcite $\delta^{18}$O values from the PDB to the SMOW scale according to Coplen et al. (1983). For the aragonitic *Pisidium* shells, we assume a temperature dependence of the $\delta^{18}$O like for inorganic calcite (Grossman and Ku, 1986; Kim et al., 2007), and the species-specific offset correction accounts for the offset between the shell carbonate and inorganic calcite.

# 3   Results

## 3.1   Isotopic composition of water

The $\delta^{18}$O of monthly precipitation at the Bredkälen GNIP station averaged over the period of 1975-1980 follows closely the annual cycle of air temperature. Values of individual months range between -4.9‰ and -19.9‰ (Figure 3b). Although the





precipitation isotope time series and the local water isotope data were collected during different periods, some relationships

can be inferred from the distributions of the isotope values.

The $\delta D$ and $\delta^{18}O$ of water from the tributary of Locknesjön plot close to the Global Meteoric Water Line (GMWL). The seasonal range of $\delta^{18}O$ in the tributary (-13.11‰ to -11.57‰) is reduced compared to the range in precipitation (Figure 4b), which indicates a substantial groundwater contribution to the stream flow. Still, a clear shift to higher $\delta^{18}O$ and $\delta D$ values in the tributary shows a marked influence of precipitation to river supply to the lake during the warm season. The isotopic values

of the groundwater inflow are more negative than all values of the tributary, and the $\delta^{18}O$ is close to the amount-weighted mean of precipitation for 1976-1980 (-13.5‰), which as expected indicates a negligible influence of evaporation.

Water samples from the lake and its outflow, on the other hand, show effects of evaporative enrichment in the heavy isotopes compared to groundwater and the tributary by about 3‰ for $\delta^{18}O$ and 17‰ for $\delta D$. This enrichment is seen in samples from all investigated years and in all seasons. The inferred local evaporation line with a slope of 4.9 and an intercept of 29‰ is

typical for lakes with evaporative water loss (Craig and Gordon, 1965; Craig, 1961).

The difference between lake water samples taken in winter and summer is small, with values ranging from -9.66‰ to -9.24‰ for $\delta^{18}O$ and from -75.4‰ to 73.6‰ for $\delta D$. The exceptional meteorological conditions during the summer of 2018 did not lead to a significant effect on the $\delta^{18}O$ of lake water, e.g. through increased evaporation. The $\delta^{18}O$ of water sampled in 2018 is only about 0.1‰ higher than in other warm season water samples (Figure 4c). A potential effect of the dry and warm

summer conditions 2018 is either masked through the mixing of lake water, or could arrive in the lake with a delay. The small seasonal and inter-annual changes of the isotopic composition can be explained by the relatively long water residence time of over 10 years in Lake Locknesjön, caused by its large volume in relation to its limited catchment size (Figure 1b). Inter-annual variations in the isotopic composition of precipitation are only reflected in lake water $\delta^{18}O$ if they last longer than the lake water residence time (von Grafenstein and Labuhn, 2021).

Variations in $\delta^{18}O$ and $\delta D$ between 0 and 20 m water depth in April 2014 do not exceed analytical uncertainty, with a range of 0.02‰ for $\delta^{18}O$ and 0.3‰ for $\delta D$. The $\delta^{18}O$ and $\delta D$ vary only minimally between 1 m and 8 m water depth in July 2018 and do not show any trends with depth. The ranges of 0.17‰ for $\delta^{18}O$ and 0.8‰ for $\delta D$ are only slightly larger than the measurement error, indicating that the lake water is well mixed above the thermocline. Lake Locknesjön displays a distinct summer stratification with a thermocline generally below 10 m (Figure 2; Eklund, 1998). Although only one summer

temperature profile per year is available, it can be concluded that the rapid transition from ice-covered conditions in May to maximum air temperature in July, together with the effects of strong winds, leads to a rather deep and well-mixed summer epilimnion.

Lake Blektjärnen shows a 0.66‰ lower $\delta^{18}O$ and a 13.1‰ lower $\delta D$ compared to Lake Locknesjön in the summer of 2018. These measurements are in agreement with previously published isotope data from Lake Blektjärnen (Andersson et al., 2010).





### 3.2 Isotopic composition of biogenic carbonates

#### 3.2.1 *Pisidium*

The $\delta^{18}$O values of the aragonite shells of *Pisidium* samples range between -8.83‰ and -7.38‰, their $\delta^{13}$C values lie between -7.20‰ and -4.00‰ (Table 1, Figure 5a). The relatively narrow ranges can be explained by the accumulation of shell carbonate throughout their lifetime of up to three years (Holopainen and Jónasson, 1989). This averages out seasonal and inter-annual variations in the water isotopic composition and water temperature. There is no significant difference in isotope values between living specimen sampled in 2018 and the valves taken from surface sediments (i.e. which were formed during previous years), possibly because their time spans of carbonate accumulation overlap. The mean $\delta^{18}$O is the same in living and dead samples, and the mean $\delta^{13}$C differs by 0.19‰.

The previously reported vital offset on the oxygen isotopic composition of *Pisidium* shells of +0.9‰ (von Grafenstein et al., 1999b) is confirmed by this study. After correction for this offset, the *Pisidium* $\delta^{18}$O falls close to expected values in equilibrium with the lake water, as observed in other studies of mollusk shells (Apolinarska et al., 2016).

#### 3.2.2 *Chara*

Samples of calcite encrustations from *Chara hispida* in Locknesjön show the largest range in isotope values compared to other species (Table 1). This can be explained by their longevity. Since *Chara* are perennial (Martin et al., 2003), different parts of the algae are formed during different years. The growth of *Chara* is apical, i.e. calcification during one growing season is limited to the apices (Coletta et al., 2001). Each analyzed sample consisted of only one internode or branchlet, which means that multi-annual environmental variations are not reflected by individual samples. There is no difference between the isotopic compositions of encrustations from living *Chara hispida* and surface sediments (Figure 6a), indicating that the environmental influences on the $\delta^{18}$O and $\delta^{13}$C of the encrustations were similar over the past few years. The perennial growth also means that there could be an overlap in the years of growth represented in living and sub-recent samples.

*Chara* growth is restricted to the time of photosynthetic activity, i.e. the summer months in our study area (Hammarlund et al., 2002). The $\delta^{13}$C is driven by fractionation during photosynthesis, which preferentially incorporates $^{12}$C in the organic matter. Consequently, DIC in the surrounding water and, in turn, the calcitic encrustations are enriched in $^{13}$C (McConnaughey and Falk, 1991). This process is reinforced by proton pumping, leading to additional $^{13}$C enrichment in the encrustations (Hammarlund et al., 1997). After correcting *Chara hispida* $\delta^{13}$C for the vital offset of 4‰, the values are largely similar to the $\delta^{13}$C values of other biogenic carbonate samples from Locknesjön (Figure 5b). The magnitude of carbon isotope fractionation in *Chara* encrustations observed here is therefore the same as in previous studies (Hammarlund and Keen, 1994; von Grafenstein et al., 2000).

The mean $\delta^{18}$O of the *Chara hispida* encrustations in Locknesjön (-9.77‰) is close that of inorganic calcite that would be precipitated in isotopic equilibrium with lake water, which has a mean $\delta^{18}$O (-9.40‰). Other studies, on the contrary, have found significant offsets from equilibrium due to kinetic effects during fast calcite precipitation (Apolinarska et al., 2016; Andrews et al., 2004).





Fine calcite $\delta^{18}O$ values fall within the range of $\delta^{18}O$ of the encrustations, but their mean is higher than the mean of the encrustations. The observed $\delta^{13}C$-values are similar for fine calcite and encrustations, which indicates that the matrix

of the carbonate-rich sediments of Lake Locknesjön is composed of disaggregated *Chara hispida* encrustation rather than allochthonous carbonates.

The two *Chara* species sampled in Lake Blektjärnen differ in the isotopic composition of their encrustations (Figure 6b). The average $\delta^{13}C$ in *Chara hispida* is significantly higher (1.04‰) than in *Chara aspera*. Such differences can be caused by habitat conditions such as stand density and by metabolic effects (Pronin et al., 2018; Coletta et al., 2001). The $\delta^{13}C$ is

driven by photosynthesis. *Chara hispida* grows in dense swards, which limits the water exchange and therefore the removal of DIC enriched in $^{13}C$ due to photosynthesis, leading to a corresponding enrichment of $^{13}C$ of the encrustations (Coletta et al., 2001). The average difference in $\delta^{18}O$ (0.46‰) is not significant. The $\delta^{18}O$ of *Chara* is driven by water $\delta^{18}O$, which leads to a corresponding, species-independent $\delta^{18}O$ of encrustations formed in the same water.

Internodes and branchlets show small differences in their mean isotope values, but these differences are not consistent

between species and lakes. In living *Chara hispida* from Locknesjön, the mean $\delta^{18}O$ of branchlets is 0.56‰ higher and the mean $\delta^{13}C$ is not significantly different from internodes. In samples from Blektjärnen, internodes and branchlets are not significantly different in *Chara aspera* and in $\delta^{18}O$ of *Chara hispida*, whereas the $\delta^{13}C$ of *Chara hispida* is higher on average in the internodes. Consequently, it does not seem to be necessary to distinguish between these parts of the alga in isotopic analyses of the encrustations.

### 3.2.3   Ostracods

The $\delta^{18}O$ values of ostracod valves range between -7.55‰ and -4.13‰, and the $\delta^{13}C$ values range between -6.90‰ and -2.08‰ (Table 1). There is no significant difference between the $\delta^{18}O$ values of *Candona candida* and *C. neglecta*. The difference in $\delta^{13}C$ is statistically significant, but the mean $\delta^{13}C$ is only 0.67‰ lower for *C. neglecta*. This general similarity in the isotopic composition of the two species is in agreement with previous studies (von Grafenstein et al., 1999b). With known

vital offsets subtracted (2.2‰ for both species; von Grafenstein et al., 1999b), the $\delta^{18}O$ values of are close to the values of other biogenic carbonates from Locknesjön (Figure 5b).

Comparing different developmental stages of the ostracods, there is a clear decrease in the average $\delta^{18}O$ values from adult valves to A-1 to A-2/A-3/A-4 instars, although the values of individual samples partly overlap (Figure 7). The timing of the molting of ostracods is controlled by temperature. In southern Germany, it has been observed that co-existing *Candona* instars

A-4 to A-2 form their valves during early to mid summer, A-1 at the onset of autumn cooling and adults between late autumn and early spring (von Grafenstein et al., 1999b). Consequently, seasonal changes in water temperature and water $\delta^{18}O$ have an influence on the $\delta^{18}O$ of the valves formed during different times of the year. Higher water temperatures lead to reduced fractionation and a decrease in the $\delta^{18}O$ of the precipitated carbonate.

As the instars A-2 to A-4 form at the time of maximum water temperature in summer, the mean $\delta^{18}O$ of their valves is

significantly lower than in A-1 and adult samples. There is no significant difference between A-2, A-3 and A-4 instars. The average $\delta^{18}O$ of A-1 samples is higher than in younger instars because A-1 instars develop when water temperature starts to





decrease in autumn. Moreover, by the end of the summer the lake is thermally stratified and water in the epilimnion is more enriched in $^{18}$O by evaporation compared to the time when the younger instars calcify their valves, leading to a further increase in the $\delta^{18}$O of A-1 valves. The valves of adult ostracods start forming in late autumn. When overturning occurs in the lake,

stratification ceases and the isotopically enriched surface water is mixed with deep water with a lower $\delta^{18}$O. Nevertheless, adult ostracods have significantly higher $\delta^{18}$O values on average compared to juvenile instars, because their valves are formed at the lowest water temperature. However, the lowest $\delta^{18}$O value of all ostracod samples was recorded in an adult specimen from 1 m water depth.

    The patterns of average $\delta^{13}$C values of ostracod valves from different instars are less clear compared to $\delta^{18}$O. The average

$\delta^{13}$C of all juvenile instars are generally similar, whereas the average $\delta^{13}$C of adult ostracods is slightly lower. The highest values, however, were observed in samples of A-1 instars. The $\delta^{13}$C of ostracod valves from exobenthic species like *Candona* depends on the carbon isotopic composition of DIC in the surrounding water. The higher $\delta^{13}$C of juvenile ostracods suggests that DIC is slightly enriched in $^{13}$C as a result of *Chara* photosynthesis and other primary productivity during the summer when juveniles calcify their valves.

## 4   Discussion

The sources of oxygen and carbon for calcifying lacustrine organisms are lake water and DIC, respectively. The isotopic composition of the biogenic carbonates is dependent on the (variable) isotopic composition of these sources, but additional factors lead to offsets in the species-specific isotopic composition compared to the compositions of water and DIC. The following sections discuss in how far these influences can be constrained though our "snapshot" approach by comparing different species,

and which paleoclimate information can potentially be obtained from the carbonates.

### 4.1   Climate signal in the $\delta^{18}$O of lake water

In order to interpret the oxygen isotopic composition of lactustrine biogenic carbonates, it is important to understand the causes of variations in the $\delta^{18}$O of the oxygen source, i.e. the lake water. Generally, lake water $\delta^{18}$O depends on the water balance, as the $\delta^{18}$O of the water input (precipitation, potentially modified though processes in the catchment) can be increased through

evaporation in the lake (Mayr et al., 2007; von Grafenstein and Labuhn, 2021). The measured values of river and groundwater inflow into Lake Locknesjön show that seasonal variations in the isotopic composition of precipitation are smoothed through groundwater supply and long lake water residence time. The difference between tributary and lake water indicates evaporative enrichment in $^{18}$O. In the absence of long-term monitoring, it is not possible to disentangle the relative importance of precipitation $\delta^{18}$O or evaporative enrichment for the $\delta^{18}$O of the lake water. Furthermore, we cannot deduce past variations in either

precipitation $\delta^{18}$O or evaporative enrichment from biogenic carbonates in lake sediments. However, since both factors increase with increasing temperature, lake water $\delta^{18}$O should contain a signal of multi-annual temperature variability. As shown by our lake water $\delta^{18}$O and water temperature profiles (Figure 2a, Figure 8), the lake water is well mixed down to a depth of about 8 m, i.e. water depth is not a decisive factor across the range of sampled depths.





### 4.2 Influence of water depth on the isotopic composition of carbonates

None of the investigated species show any significant trends in average $\delta^{18}O$ or $\delta^{13}C$, nor in the range of isotope values, along the transect from 1 to 8 m water depth (Figure 8). The largest temperature changes can be expected in shallow water, which would lead to larger ranges of $\delta^{18}O$ values at the shallowest sampling depths. However, since no systematic changes were observed in the carbonate isotopic compositions, nor in the $\delta^{18}O$ of the lake water, we conclude that the calcification conditions and the isotopic compositions of the carbon and oxygen sources did not vary significantly within the sampled range

of water depths. The effects of water temperature changes and evaporative enrichment in $^{18}O$ of lake water are the same, and the water at 8 m depth appears to be well-mixed and clearly above the summer thermocline for the time span represented in the living and sub-recent specimens.

### 4.3 Causes of inter-specific differences in the isotopic composition of carbonates

Vital offsets are the major cause of differences in the oxygen isotopic composition between species (Table 1, Figure 5a).

With these known and constant offsets corrected for, the mean $\delta^{18}O$ values of all species analyzed in our study plot closer to each other and their ranges show larger overlaps (Figure 5b). The remaining differences can be explained by environmental influences, which vary with the timing and length of the respective calcification periods. The mean values of each species depend on the time of the year when the carbonate is precipitated, involving potentially large differences in water temperature. The range of values of a species depends on the length of the calcification period of each sample, a longer period smoothing

seasonal and inter-annual variability.

The mean $\delta^{18}O$ value of *Chara hispida* encrustations lies clearly below the mean $\delta^{18}O$ values of all other species. The mean $\delta^{18}O$ values of *Pisidium* and juvenile ostracods are higher, and adult ostracods show the highest mean. The timing of calcification in *Chara hispida* is limited to periods of maximum photosynthetic activity in summer, which can last only a few weeks. These periods are likely also the periods of highest temperatures, which would explain the low $\delta^{18}O$ of *Chara hispida*

encrustations compared to other carbonates. Seasonal temperature variations also cause the differences observed between adult and juvenile ostracods (see Sect. 3.2.3). As juvenile ostracods precipitate calcite mainly during the summer months, their values overlap with values of *Chara hispida* encrustations. The calcification period of *Pisidium* is the longest of all studied organisms and lasts throughout the ice-free period from May to November (Andersson et al., 2010). Most calcification occurs during summer, whereas their growth is substantially suppressed during winter (von Grafenstein et al., 1999b). The mean

$\delta^{18}O$ value of *Pisidium* is slightly lower than that observed for juvenile ostracods A-2,3,4, although the former are exposed to lower average water temperatures during their calcification period and thus a higher carbonate $\delta^{18}O$ would be expected due to temperature-dependent fractionation. The lower Pisidium $\delta^{18}O$ values may reflect the influence of relatively $^{18}O$-depleted snow-melt supply to the lake in spring and from mixing of surface water with $^{18}O$-depleted bottom water in autumn, processes that juvenile ostracods are not exposed to.

The range of $\delta^{18}O$ is smallest for *Pisidium* (1.45‰) and largest for *Chara hispida* (4.28‰). Although *Chara* are perennial, the samples measured in this study, encrustations from single internodes or branchlets, were precipitated during a single

 

growing season, i.e. individual samples may represent different years, hence explaining their larger isotopic range. The average lifetime of *Pisidium* of 3 years, together with their long annual calcification period probably explain their relatively narrow range of $\delta^{18}$O.

Overall, seasonal water temperature changes seem to be the dominant cause of differences in the $\delta^{18}$O of the investigated biogenic carbonates (after vital offset correction). The $\delta^{18}$O of the lake water, on the contrary, does not have a strong influence due to the small seasonal and inter-annual variability around the mean of -9.53‰ during the years of observation.

The carbon isotopic composition of lacustrine biogenic carbonates depends primarily on the $\delta^{13}$C of DIC in lake water. Observations of $\delta^{13}$C of DIC in groundwater or lake water are not available for Lake Locknesjön but some general assump-
tions can be made. The $\delta^{13}$C of groundwater DIC is dependent on the open/closed system conditions, which determine the relative contributions from bedrock carbonates and soil air $CO_2$ (Clark and Fritz, 1997). The $\delta^{13}$C of carbonate bedrock in the catchment, i.e. resurge deposits in the Lockne Crater and post-impact sediments, is close to -0.5‰ (Ormö et al., 2010). The $\delta^{13}$C of microbially respired soil air $CO_2$ reflects that of the local vegetation, which can be estimated to be around -25‰ based on tree ring cellulose data from the study area (unpublished data). It is increased by ~4‰ through fractionation by
diffusion as $CO_2$ outgasses along the $p$CO$_2$ gradient between the soil and the atmosphere (Cerling et al., 1991). As a result of the well-mixed character of groundwater recharging the lake as evidenced by water isotope monitoring data (Figure 4), likely reflecting a relatively long residence time, seasonal variations in $\delta^{13}$C of groundwater DIC are probably small or absent.

On the contrary, DIC in lake water usually shows substantial seasonal variations, increasing in the epilimnion during the stratified warm period and decreasing to the lake-wide mean $\delta^{13}$C during the holomixis (Decrouy et al., 2011a; von Grafen-
stein et al., 1999b). In Lake Locknesjön, warm-season increases in $\delta^{13}$C are most likely mainly induced by the photosynthetic activity of *Chara* (*cf.* Sect. 3.2.2), which can be assumed to also raise the mean $\delta^{13}$C of lake water DIC compared to groundwater.

*Chara hispida* encrustations exhibit a significantly higher $\delta^{13}$C than ostracod and *Pisidium* valves (Figure 5a). While the $\delta^{13}$C of *Pisidium* and ostracod calcite reflects the $\delta^{13}$C of DIC, the $\delta^{13}$C of *Chara hispida* encrustations is influenced by
fractionation between DIC and the organic matter produced during photosynthesis. Juvenile *Candona* show a slightly higher $\delta^{13}$C than adult *Candona* and *Pisidium*, which suggests that DIC is slightly enriched in $^{13}$C by *Chara* photosynthesis in summer when juvenile ostracods calcify their valves (Figure 7). However, this effect on juvenile ostracods is much weaker than on the carbonate precipitated at the site of photosynthesis, i.e. in the *Chara hispida* encrustations. No other effects related to seasonal changes in $\delta^{13}$C of DIC or lake water temperature on the $\delta^{13}$C of biogenic carbonates can be inferred from our
data.

Although probably minor in the carbonate-rich and therefore well-buffered system of Lake Locknesjön, seasonal variations in lake-water pH may contribute to the observed inter-specific differences in $\delta^{13}$C and $\delta^{18}$O. The pH measured at depths above 8 m ranges from 8.05 to 8.33, the maximum occurring in August. In cultured foraminifera, a slight lowering of $\delta^{18}$O and $\delta^{13}$C has been observed in response to increasing pH (Spero et al., 1997). Field observations of sediment trap calcite and lake water
pH confirm this relationship (Teranes et al., 1999). However, further measurements of lake water pH and contemporaneously precipitated calcite would be needed to investigate the influence of pH variations on biogenic carbonates in Locknesjön.



An additional factor that could play a role for inter-specific differences is the micro-habitat of the respective species. The most prominent spatial change in micro-habitat at our sampling sites is the disappearance of *Chara* below 7 m depth. No influence of this change is evident on the carbonate isotopic compositions of the other species (Figure 8). Furthermore, groundwater
inflow may lead to local decreases in temperature and $\delta^{18}O$ of the lake water (*cf*. Sect. 2.1), but such inflows were not identified in the direct vicinity of the sampling sites, and no local decreases in carbonate $\delta^{18}O$ related to groundwater inflow were observed.

### 4.4 Uncertainties in water temperature reconstruction

Water temperature measurements contemporaneous with the formation of the analyzed biogenic carbonate samples are not
available. Nevertheless, the range of observed seasonal water temperature changes in previous years (Figure 2) is consistent with the reconstructed water temperature ranges based on mean lake water $\delta^{18}O$ and mean carbonates $\delta^{18}O$ for the respective species (Figure 9). The highest reconstructed temperatures were obtained from *Chara hispida* calcifying in summer and the lowest temperatures from adult *Candona* calcifying in winter. This observation confirms that the differences in carbonate $\delta^{18}O$ between species (after correcting for vital offsets) are driven by varying water temperatures during their respective periods of
calcification.

For a given calcite $\delta^{18}O$ value, reconstructed temperatures calculated based on minimum and maximum observed lake water $\delta^{18}O$ values yield a range of 1.9 °C (Figure 9). The variability in $\delta^{18}O$ between individual carbonate samples, on the contrary, has a larger influence on the reconstructed temperature than the variability in lake water $\delta^{18}O$. Temperatures calculated based on the mean $\delta^{18}O$ of lake water samples and the minimum and maximum values of carbonates from the respective species
yield temperature ranges of 3.6 °C for juvenile *Candona candida*, >5 °C for all other ostracod instars, and 21.0 °C for *Chara hispida* (Figure 9). The lowest reconstructed water temperatures, near 0 °C for adult *Candona*, match the lowest observed lake water temperatures at Lake Locknesjön. The maximum reconstructed water temperature of 29 °C for *Chara hispida*, however, is rarely reached in maximum daily air temperatures in the study area, and therefore unlikely to occur in the open water column.

A part of these large ranges in reconstructed temperatures are probably related to temporary variations in shallow water
temperature that greatly exceed those indicated by the observed annual minimum and maximum values (Figure 2). In particular, calcification of *Chara* encrustations may take place in secluded micro-habitats at anomalously high water temperatures in summer, potentially also involving oxygen-isotope disequilibrium effects (Pentecost et al., 2006). In addition, deviations from normal seasonal calcification patterns may occur. For example, summer generations of adult ostracods have been observed, likely as a consequence of thunderstorms, leading to temporary cooling of surface water (von Grafenstein et al., 1999b).

### 4.5 Implications for paleoclimate studies

Our data exhibit relatively large isotopic variability within one species (Figure 5a), based on samples collected either from living organisms of from surface sediments, i.e. representing calcification during a time interval of a few years in the recent past. Analytical techniques allow the analysis of small quantities of calcite corresponding to e.g. a few adult ostracod valves or a single stalk of a *Chara* encrustation. However, our results demonstrate that isotopic records obtained from biogenic carbonates





in lake sediments for the study of past changes in environment and climate should be based on either (i) sufficiently large numbers of individual samples at each analyzed sediment interval, or (ii) homogenized samples incorporating sufficiently large numbers of specimens from each sediment interval, in order to obtain representative average values for the corresponding time spans.

The observed differences between our investigated species highlight the importance of performing isotopic measurements
on well-identified remains of specific biogenic carbonate components in lake sediments rather than on bulk carbonate. Isotope records obtained on bulk sediment may reflect changes in the relative contributions of carbonate from different species and variations in detrital carbonate deposition that are not directly related to the environmental variables under study, thus adding complexity to interpretations (Hammarlund and Buchardt, 1996). For example, *Chara* encrustations are clearly the dominant component of the coarse fraction of the surface sediments of Locknesjön at the sampled locations at water depths of up to 8 m.
Changes in lake level at a coring site could lead to disappearance of *Chara* if wave and ice action (in shallow water) or lack of light (in deep water) hinder their growth, which would shift bulk carbonate $\delta^{18}O$ to higher values. If bulk carbonate isotope records are complemented with contemporary monitoring data and proper characterization of the sediment composition, their cautious interpretation may still yield important information. However, the isotopic analysis of specific carbonate components, preferably involving multiple taxa, can provide more detailed insight into past changes in climate because of their differences
in the timing of calcification, and consequently water temperature and water $\delta^{18}O$.

Past changes in water depth, caused by sediment accumulation and compaction or hydrological changes, may also influence isotopic records by affecting bottom water temperatures independently of other changes in climate. However, at Lake Locknesjön, potential changes in water depth in the range of 1 to 8 m do not seem to exert any significant influence on the isotopic composition of biogenic carbonates under modern conditions. We can expect that isotope proxy records from sediment cores at
this site remain unaffected by possible paleo-water depth changes in this range, as long as the sediment surface remains above the summer thermocline.

## 5 Conclusions

This study presents a modern snapshot of the variability of the carbon and oxygen isotopic compositions of lacustrine biogenic carbonates from living organism and sub-recent sediments in Lake Locknesjön, which allows us to draw the following
conclusions.

(1) We constrained the controls on the isotopic composition of biogenic carbonates in Locknesjön. The large differences between species are mainly caused by species-specific vital offsets. Additional differences are caused by the timing and length of the calcification periods of the respective species, being exposed to seasonally changing water temperatures. In contrast, seasonal and inter-annual variations in lake water $\delta^{18}O$ are small, at least over the time span of observation (2013-2014 and
2018). The influence of changing lake water $\delta^{18}O$ on carbonate $\delta^{18}O$ is outweighed by the opposing effect of changing water temperature.
(2) Based on the understanding of these environmental controls, it is possible to estimate seasonal water temperature changes from the $\delta^{18}$O of lake water and of specific biogenic carbonates. This approach is particularly applicable for groundwater-fed lakes with long residence times like Lake Locknesjön, where lake water $\delta^{18}$O varies within a relatively narrow range. Under such conditions, seasonal changes in lake water temperature are clearly reflected in the $\delta^{18}$O of biogenic carbonates through their effect on isotope fractionation between water and carbonate. On longer (decadal to millennial) time scales, significant changes in air temperature can be expected. The consequent water temperature changes could still exert the dominant control on carbonate $\delta^{18}$O. However, the effects of changes in $\delta^{18}$O of precipitation and groundwater, brought about by related or independent climate dynamics, as well as potential variations in evaporative enrichment in the $\delta^{18}$O of the lake water (changing evaporation/inflow ratio of the lake) also need to be considered in paleoclimate studies.

(3) Our results have implications for the design of subsampling strategies aiming at reconstructions of past changes in environment and climate based on isotopic analyses of sediment cores. The intra-specific variability in $\delta^{18}$O and $\delta^{13}$C of biogenic carbonates highlights that care must be taken to obtain representative subsamples of a species for each time interval. The inter-specific variability observed in Lake Locknesjön adds to the growing body of evidence demonstrating the importance of performing measurements on specific carbonate components of lake sediments rather than on bulk carbonate. Sampling strategies involving multiple specific biogenic carbonate components are likely to produce more detailed and comprehensive environmental reconstructions.

*Data availability.* Water and carbonate isotope data obtained in this study will be published in the Pangaea database upon acceptance of the manuscript.

*Video supplement.* A video showing the movement of sediment on the lake floor at a groundwater inflow is available at this link.

*Author contributions.* IL and UvG designed the study and coordinated the field work. Samples were prepared by FT and UvG. Water isotope measurements were performed by BM, carbonate isotope measurements by HK. FT and IL created the figures. IL and FT wrote the paper, with contributions from DH, UvG and HK. All co-authors contributed to the interpretation and discussion of the results. DH and UvG were principal investigators of the project within which this study was conducted.

*Competing interests.* The authors declare that they have no conflict of interest.





*Acknowledgements.* This study would not have been possible without the scientific diving team of Benedikt Beck, Matthias Galm, Jan Leidholdt and Stefan Zimmermann. The authors are grateful to Östersund Diving School for arranging the diving equipment; to Malin Bernhardsson at the County Administration Board of Jämtland for providing a boat as well as water temperature and pH data from the Triple Lakes project; to Anna Eklund at the Swedish Meteorological and Hydrological Institute for providing archived water temperature data; and to Sten-Åke Jonsson for collecting water samples. Wolfgang Bevern, Birgit Meyer and Maike Steinkamp are thanked for helping with the carbonate isotope measurements. This study was funded by the Swedish Research Council as part of the French-Swedish Common Research and Training Programme on Climate and the Environment (grant no. 349-2012-6291) and by the Central Research Development Fund of the University of Bremen (grant no. 2018/ZF08/FB8/Labuhn). The article processing charges for this open-access publication were covered by the University of Bremen.




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

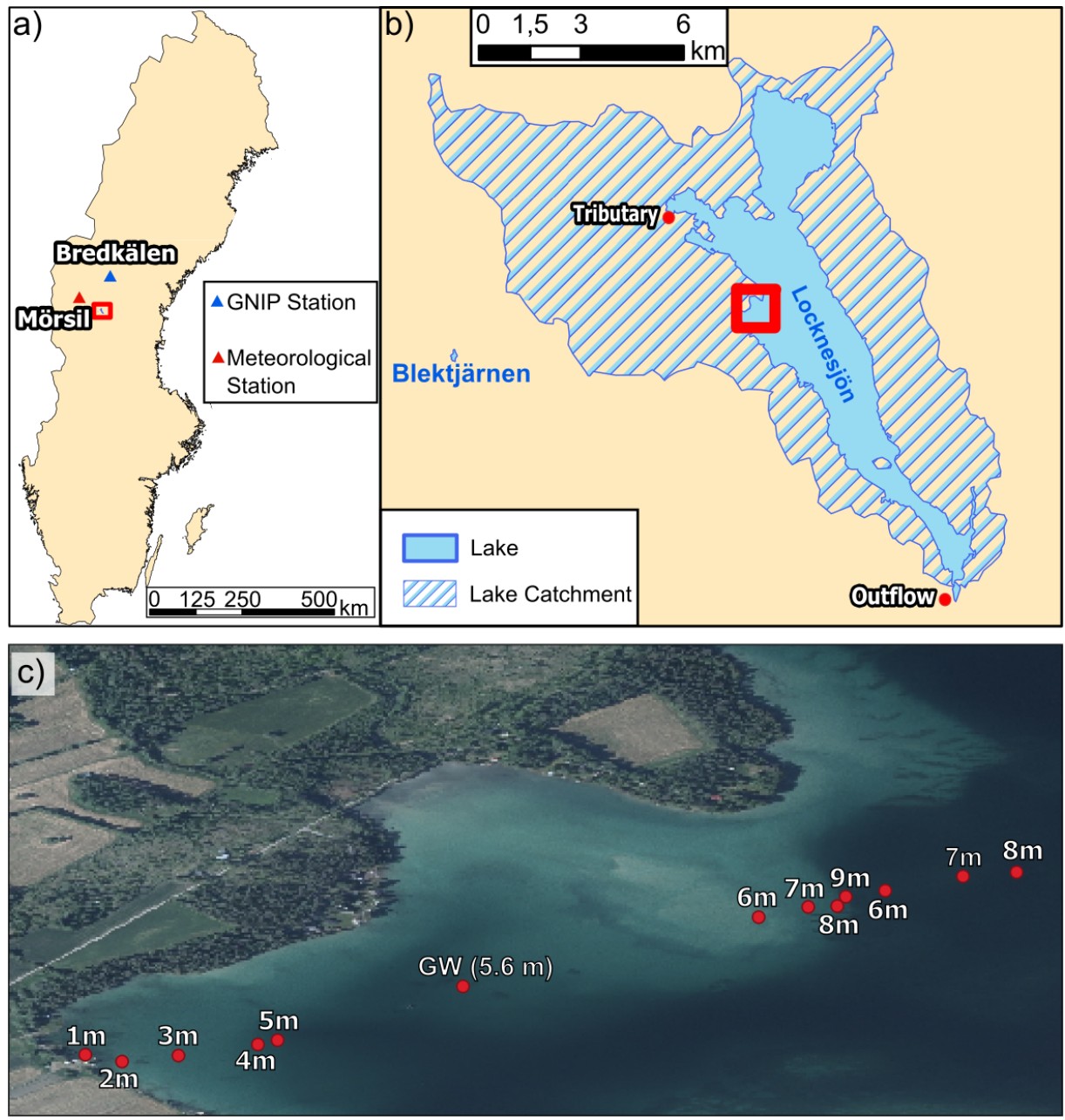

**Figure 1.** a) Map of Sweden showing the location of the study area (red rectangle), of the GNIP station Bredkälen and of the meteorological station Mörsil. b) Lake Locknesjön with its catchment (hatched) and Lake Blektjärnen. c) Positions of the sampling sites on a transect from the shore towards the center of the lake. The area shown in b) is marked by the red rectangle in a) and the area shown in c) is marked in b). (Country outline from USGS, lake and catchment outline from SMHI (2020a); orthophoto © Lantmäteriet 2020).



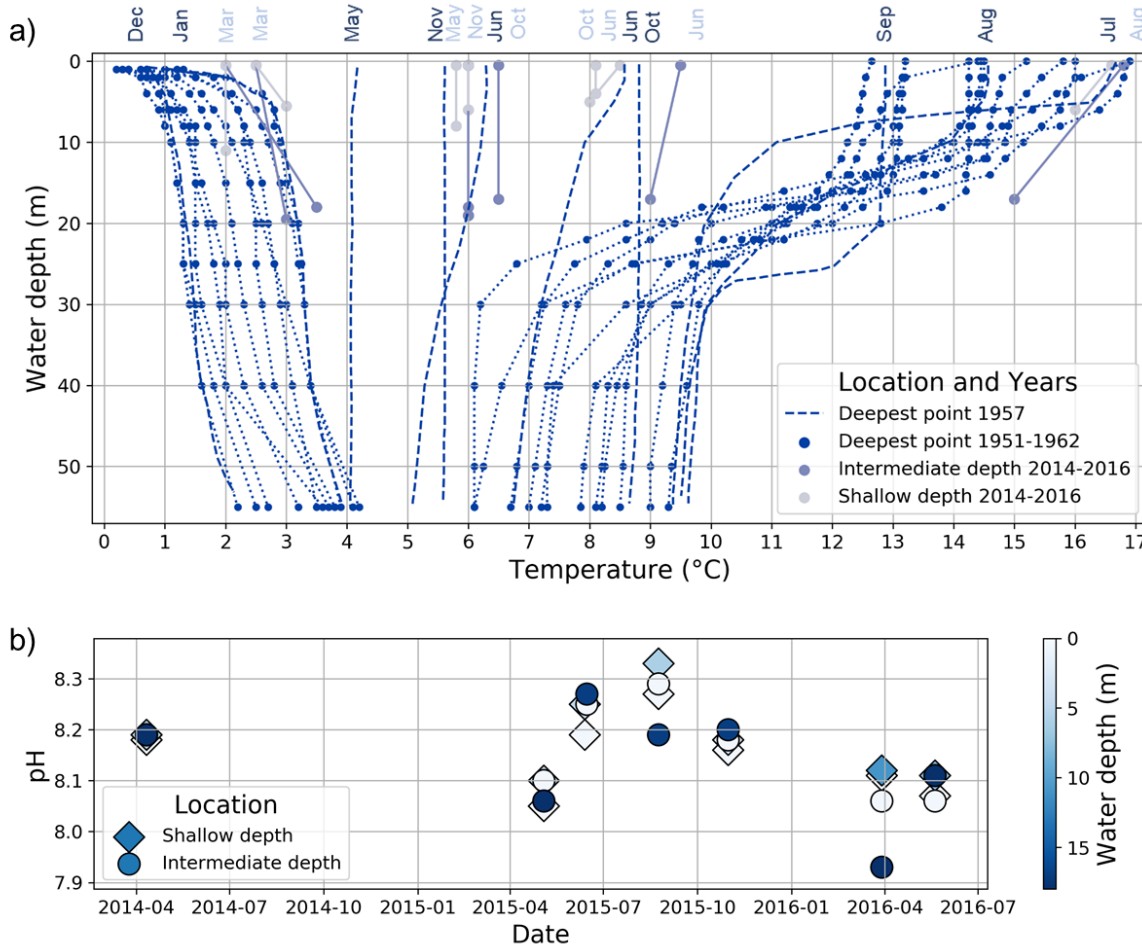

**Figure 2.** a) Water temperature profiles from Lake Locknesjön. Dark blue dots represent measurements taken between 1951 and 1962, two times per year approximately at the time when the water was coldest during winter (February-March) and warmest during summer (August). Profiles extend from the surface to the bottom at the lake's deepest point. The dashed lines show measurements taken at the same point throughout the year 1957 in the indicated months (data from Eklund, 1998). Other dots represent measurements taken between 2014 and 2016 from surface and bottom water at a two sites of shallow (< 11 m, light blue dots) and intermediate (< 20 m, light gray dots) water depth (data from the Triple Lakes Project, http://www.triplelakes.se/). Dotted and solid lines connect measurements taken from different depths at the same location and on the same day. b) pH measured in surface and bottom water at two sites of shallow and intermediate depth between 2014 and 2016 (data from the Triple Lakes Project).



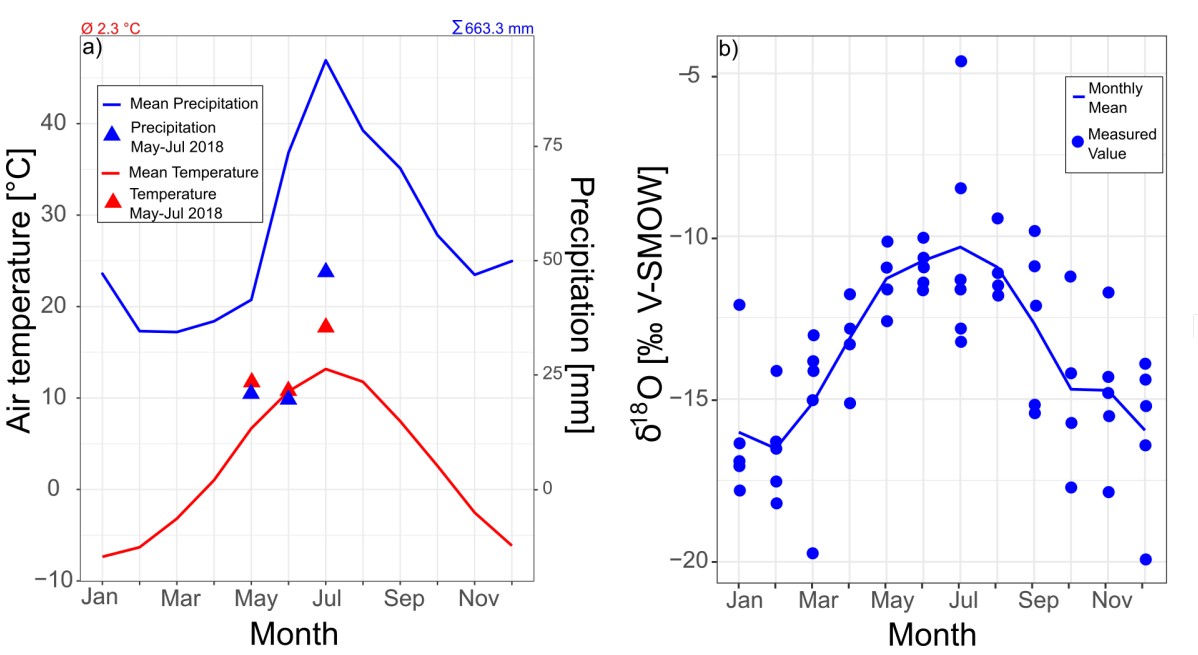

**Figure 3.** a) Monthly climatology of air temperature and precipitation in the study area from the meteorological station Mörsil for the period of 1975-2005 (data from SMHI, 2020b). The values for May, June and July in the year of sampling (2018) are indicated by the triangles. b) $\delta^{18}O$ in precipitation at the GNIP station Bredkälen for the period of 1975-1980 (data from IAEA/WMO, 2019). See Figure 1 for the locations of the stations.





**Figure 4.** Isotopic composition of water in the study area. The symbols represent different locations and years of sampling. The color scale indicates the month of sampling, with blue values representing the cold period and red values representing the warm period of the year. a) Average monthly $\delta^{18}$O of precipitation for the period of 1975-1980 at the GNIP station Bredkälen (data from IAEA/WMO, 2019, note that $\delta$D is not available for this station). The large black diamond shows the mean $\delta^{18}$O for 1976-1980 weighted by the amount of precipitation. b) $\delta$D and $\delta^{18}$O of lake water (sampled in 2013-2014 and 2018), the groundwater inflow (2018), and water in the tributary (2013-2014) and outflow (2013-2014) of Lake Locknesjön, and lake water from Lake Blektjärnen (2018). The regional meteoric water line, inferred from measurements from the tributary of the lake (RMWL; $\delta D = 7.6 * \delta^{18}O + 3.6$) is plotted in black, and a Local Evaporation Line (LEL; $\delta D = 4.9 * \delta^{18}O + 29$) estimated from the isotopic composition of Locknesjön lake water in blue. c) Zoom on the lake water isotope data from Lake Locknesjön.



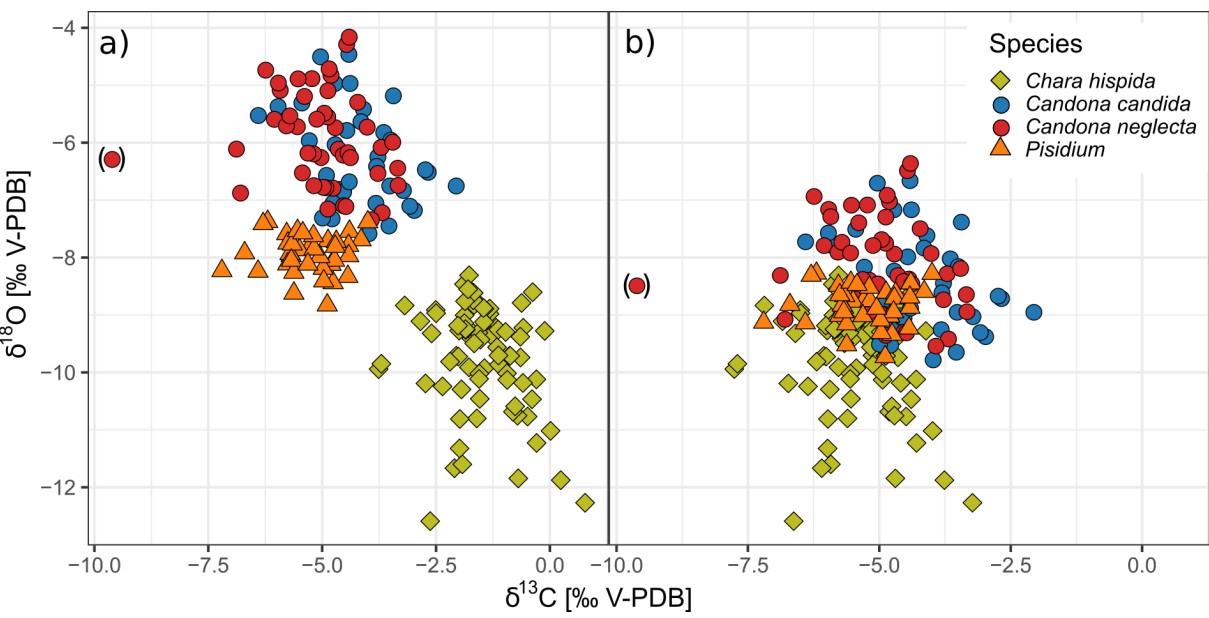

**Figure 5.** Isotopic composition of specific biogenic carbonate samples collected from surface sediments and living organisms from Lake Locknesjön. a) $\delta^{18}$O and $\delta^{13}$C of calcite encrustations from *Chara hispida* and of valves from *Candona candida*, *Candona neglecta* and *Pisidium*. b) The same data with published vital offsets corrected for: $\delta^{13}$C *Chara hispida*: -4‰ (value from Hammarlund and Keen, 1994), $\delta^{18}$O ostracods: -2.2‰, $\delta^{18}$O *Pisidium*: -0.9‰ (values from von Grafenstein et al., 1999b). The data point marked in brackets was identified as a major outlier and left out from further analysis and figures.





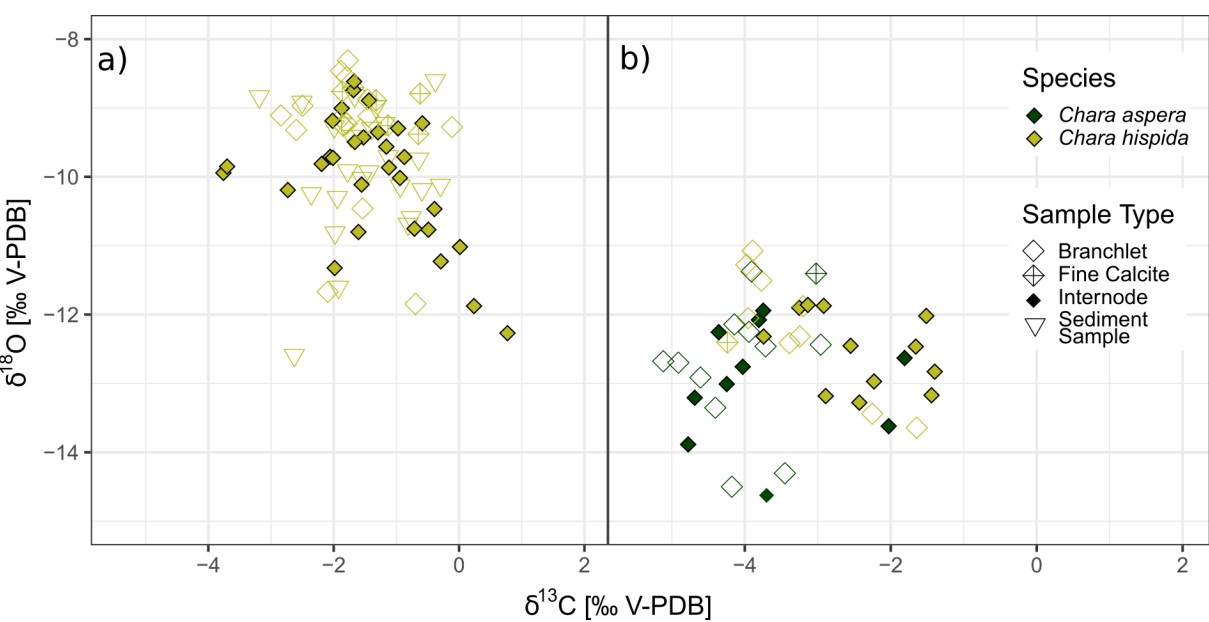

**Figure 6.** Isotopic composition of calcite encrustations from different types of *Chara* samples. Internodes, branchlets and fine calcite were collected from living *Chara*, "sediment sample" refers to fragments of *Chara* encrustations taken from surface sediments. a) $\delta^{18}$O and $\delta^{13}$C of *Chara hispida* samples from Lake Locknesjön. b) $\delta^{18}$O and $\delta^{13}$C of *Chara hispida* and *Chara aspera* samples from Lake Blektjärnen.



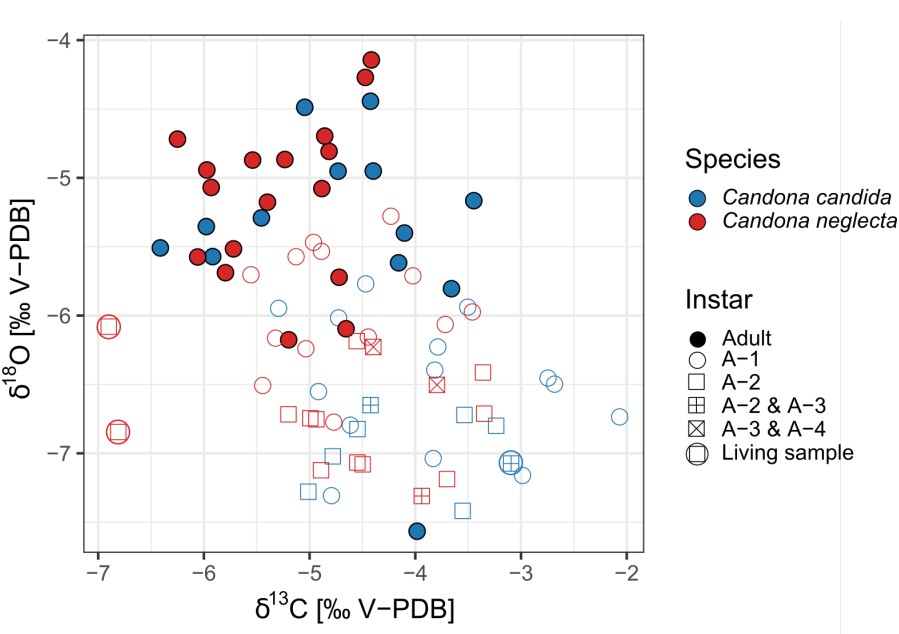

**Figure 7.** Isotopic composition ($\delta^{18}$O and $\delta^{13}$C) of the valves from two species of ostracods, *Candona candida* (blue) and *Candona neglecta* (red), taken from living specimen (encircled symbols) and sub-recent samples in the surface sediments of Lake Locknesjön. The symbols represent the instars. See the main text for explanation of the combinations of different instars.


**Figure 8.** Inter- and intra-specific variations in the isotopic composition ($\delta^{18}O$ and $\delta^{13}C$) of biogenic carbonates sampled along a transect in Lake Locknesjön at water depths from 1 to 8 m. The colors indicate the species, and the shaded areas represent the ranges of their $\delta$-values. The symbols represent different sample types (as in Figure 6 for *Chara hispida* and as in Figure 7 for ostracods). The light blue stars represent $\delta^{18}O$ of water samples taken at the same locations. The values of lake water are reported in ‰ relative to V-SMOW.





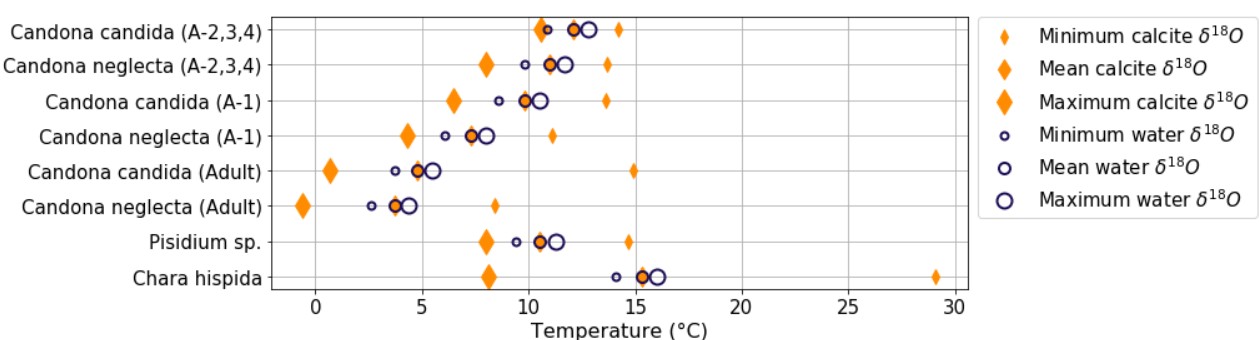

**Figure 9.** Theoretical temperature calculated from water and carbonate $\delta^{18}O$, assuming the carbonate was precipitated in isotopic equilibrium with the water (according to Kim and O'Neil, 1997). Temperatures are calculated using the mean water $\delta^{18}O$ and the range of measured carbonate $\delta^{18}O$ values (minimum, mean and maximum) for each species/instar (yellow diamonds), as well as using the mean carbonate $\delta^{18}O$ of each species/instar and the range of observed water $\delta^{18}O$ values (blue circles). The measured carbonate $\delta^{18}O$ values have been corrected for known vital offsets.





**Table 1.** Isotopic composition of specific biogenic carbonates from living organisms and from surface sediments of Lake Locknesjön. Mean, standard deviation (SD), minimum, maximum and range of $\delta^{18}$O and $\delta^{13}$C values, as well as the number of samples are presented for *Chara hispida*, *Candona candida*, *Candona neglecta* and *Pisidium sp*. All values are reported in ‰ relative to V-PDB.

|  | **Species** | **Instar** | **Mean** | **SD** | **Min** | **Max** | **Range** | **No. samples** |
|---|---|---|---|---|---|---|---|---|
| $\delta^{18}$O | *Chara hispida* |  | -9.77 | 0.94 | -12.59 | -8.31 | 4.28 | 100 |
|  | *Candona candida* | Adult | -5.38 | 0.77 | -7.55 | -4.43 | 3.12 | 13 |
|  |  | A-1 | -6.47 | 0.48 | -7.29 | -5.76 | 1.54 | 14 |
|  |  | A-2,3,4 | -6.97 | 0.27 | -7.42 | -6.65 | 0.77 | 8 |
|  | *Candona neglecta* | Adult | -5.13 | 0.58 | -6.16 | -4.13 | 2.03 | 17 |
|  |  | A-1 | -5.92 | 0.44 | -6.76 | -5.27 | 1.50 | 13 |
|  |  | A-2,3,4 | -6.73 | 0.38 | -7.31 | -6.08 | 1.23 | 16 |
|  | *Pisidium sp.* |  | -7.94 | 0.34 | -8.83 | -7.38 | 1.45 | 37 |
| $\delta^{13}$C | *Chara hispida* |  | -1.48 | 0.83 | -3.76 | 0.77 | 4.53 | 100 |
|  | *Candona candida* | Adult | -4.75 | 0.94 | -6.42 | -3.46 | 2.97 | 13 |
|  |  | A-1 | -3.88 | 0.98 | -5.30 | -2.08 | 3.23 | 14 |
|  |  | A-2,3,4 | -4.02 | 0.75 | -5.01 | -3.09 | 1.92 | 8 |
|  | *Candona neglecta* | Adult | -5.30 | 0.59 | -6.26 | -4.42 | 1.83 | 17 |
|  |  | A-1 | -4.70 | 0.67 | -5.56 | -3.47 | 2.09 | 13 |
|  |  | A-2,3,4 | -4.66 | 1.07 | -6.90 | -3.35 | 3.55 | 16 |
|  | *Pisidium sp.* |  | -5.29 | 0.72 | -7.20 | -4.00 | 3.20 | 37 |