# Peer review of "A modern snapshot of the isotopic composition of lacustrine biogenic carbonates – Records of seasonal water temperature variability"

_Biogeosciences, 2021_

## Author Comment (AC1)

**A modern snapshot of the isotopic composition of lacustrine biogenic carbonates – Records of seasonal water temperature variability**

Inga Labuhn, Franziska Tell, Ulrich von Grafenstein, Dan Hammarlund, Henning Kuhnert, and Bénédicte Minster

**RC1**

Dear Authors,

I have very much enjoyed reading your work. The contribution which looks back on what material we actually analyse, and on what are the inherent sample limitations is valuable and timely, in particularly now, when technological advances allow for more precise and more sophisticated measurements. The paper is informative and generally well written, and Biogeosciences is a most adequate venue for this work.

I have several general minor-to-moderate comments which (I hope) will improve the readability and the reception of the manuscript.

We thank Reviewer 1 for her valuable and constructive feedback! Our responses and explanations of the modifications in the manuscript are inserted below after each comment (in blue).

The language – please try to be as specific and consistent as possible. Dealing with isotopes and environmental controls, the vocabulary can be daunting, especially for less familiar readers. Please, when talking about 'precipitation' note each time if you refer to atmospheric (rainfall) or carbonate precipitation. Also, perhaps it is worth to explain once and upfront (but not as in the present version in the abstract) all the environmental factors influencing isotopic composition of carbonates and their direction. As of yet, provided explanation is correct but condensed to two long and complex sentences in the abstract. Again, please keep in mind readers less familiar with principles of stable isotope geochemistry and shrieking when 'fractionation' is mentioned. The fact that oxygen isotope fractionation is temperature-dependant, but the process happens (1) in the atmosphere and (2) in the ambient water, and drives the isotopic composition of water/ carbonate in two different directions is probably best explained using a simple sketch? I do agree that a picture is worth a thousand words, and in this case a well-designed but simple figure could improve the clarification of processes influencing d18O in lacustrine carbonates. Such figure would be a great asset in the introduction. Shall you decide to leave out the sketch option, please explain the processes consequently starting with atmospheric temperature effect on rainfall oxygen composition and lake water composition (additionally through evaporation) and only then

move to ambient water temperature influence on carbonate precipitation (modified by vital offsets).

We have carefully revised the manuscript with respect to language, and paid attention to specify whether we talk about atmospheric precipitation or carbonate precipitation.

The sentences explaining the factors influencing the isotopic composition of carbonate have been removed from the abstract. We have incorporated this explanation in the introduction instead. As suggested, we have added the following figure to illustrate the different influencing factors on the d18O of lacustrine biogenic carbonates to highlight the opposite effects of temperature on lake water d18O and on fractionation (Fig. 1 in the revised manuscript).

[Figure]

**Figure 1:** Schematic representation of the influences on the oxygen isotopic composition (d18O) of lacustrine biogenic carbonates. The isotopic composition of the water, which is dependent on air temperature as indicated in ①, is reflected in the d18O of the carbonates. There are additional influences on lake water d18O, such as catchment and lake hydrology and the seasonal distribution of precipitation, which are not represented

in the figure. The fractionation of oxygen isotopes between water
and carbonate is also temperature dependent, as indicated in ②.
Note the opposite influence of temperature on carbonate d18O in
① and in ②. Lastly, "vital effects" which are dependent on the
specific physiology of each species, lead to a constant offset
between the d18O of the biogenic carbonate and inorganic
carbonate which would precipitate in isotopic equilibrium with
the water.

In the chapter 'Material and methods' the 'material' is actually not described. An SEM
image of Candona, an SEM or macro image of Chara elements and perhaps a macro
image of Pisidium would be a good addition. Also, I would welcome a sketch of Chara
components (branchlet and internote) as I am familiar mostly with oospores and it took
me a while to understand what to you refer to as 'encrustation'.

We have added macro images in Figure 2 to provide examples of the different sample
types. The figure now also includes a photograph of a living Chara where branchlets and
internodes are indicated. The following sentence has been added to the "Material and
methods" section (Lines 167-169):

"The following biogenic carbonate components were obtained from
the sampling and prepared for stable isotope measurements:
encrustations around the internodes and branchlets of Chara,
valves of Pisidium, and valves of juvenile and adult Candona
candida and Candona neglecta (see examples in Figure 2)."

[Figure]

**Figure 2:** a) Map of Sweden showing the location of the study area (red rectangle), of the GNIP station Bredkälen and of the meteorological station Mörsil. b) Lake Locknesjön with its catchment (hatched) and Lake Blektjärnen. c) Positions of the sampling sites on a transect from the shore towards the center of the lake. The area shown in b) is marked by the red rectangle in a) and the area shown in c) is marked in b) (Country outline from USGS, lake and catchment outline from SMHI 2020; orthophoto © Lantmäteriet 2020). d) Photograph of a Chara hispida in Lake Locknesjön. An internode and a branchlet are indicated on the image. The part of the Chara hispida visible on the photograph measures about 2 cm in height. e) Images of different lacustrine biogenic carbonates analyzed in this study: (left) Pisidium

valve, (middle) adult Candona candida valve, (right) Chara
hispida encrustation.

Field sampling. I wish to see a more detailed information on field sampling. How do one take a less than 1 cm surface sediment (with a small shovel) from a water depth of more than 1 m? I imagine that one needs to employ a diver? How was the water sampling in 2013 and 2014 done? With Niskin Bottles? How was the Chara sampled?

We thank the reviewer for pointing out that some details on the sampling procedures were missing. Yes, the samples were taken by divers. This information was given already in the first submission, but we have rephrased the sentence and put it more prominently at the beginning of the paragraph to make it clear.

Water samples up to 8 m depth were taken directly into small glass bottles by the divers. Water samples down to 20 m depth (the 2013-2014 samples) were taken with a UWITEC gravity corer. Samples from the tributary and outflow were taken from the shore into small glass bottles. The samples of living Chara were hand picked by the divers. We have added this information in Section 2.2 (Lines 154-165).

"In July 2018, a diving team was employed to collect samples of
lake water, surface sediments and living Chara hispida. [...]
Surface sediment samples of about 8x8 cm2 and less than 1 cm
depth were taken with a small shovel and filled into sealable
plastic bags. Water samples were taken directly into small glass
bottles.

Living Chara hispida were only found down to 7 m water depth. Up
to six individuals were sampled at each location cutting off the
whole algae by hand. [...]

Further water samples were taken in 2013 and 2014 at depths from
0 to 20 m at Lake Locknesjön (11 samples), at one of its main
tributaries (Musån, 13 samples), and at its outflow (Forsaån, 6
samples). Water samples down to 20 m depth were taken with a
UWITEC gravity corer, the tributary and outflow samples were
taken from the shore into small glass bottles."

I see no justification for sampling Lake Blaktjärnen – its Chara results are not well incorporated into the rest of the paper. Please, if you want to keep them make sure that the reader knows why they are relevant and how they fit into the general picture.

The two lakes - Blektjärnen and Locknesjön - are difficult to compare because of their different size, depth and hydrology. Our motivation for sampling Lake Blektjärnen was to compare the isotopic composition of two different Chara species from the same site, as only one species was observed in Locknesjön during our field campaign. We found a significant difference in d13C between species that can be attributed to their metabolism and habitat preferences. We understand that this result seemed to be disconnected from the general picture of the manuscript. The conclusion that may be drawn from the Blektjärnen result is that we must test if we can combine carbonates on the genus level (as appears to be the case for Candona candida and Candona neglecta in Locknesjön),

or if the species level must be measured separately (as appears to be the case for Chara hispida and Chara aspera in Blektjärnen). This implies that the species level must be identified in fossil carbonate remains for paleoclimate studies. To better incorporate this result into the big picture, we have added the following sentences in the discussion of the causes of inter-specific differences in the isotopic composition of carbonates due to micro-habitat conditions (Lines 434-435):

"On the contrary, the example of Lake Blektjärnen shows that the d13C of different Chara species can be influenced by their micro-habitat, and that consequently a distinction at the species level is necessary for the interpretation of isotope records."

I feel awkward promoting my own work, but you may want to refer to the papers by McCormack et al., 2018 and McCormack & Kwiecien 2021; the most recent component-specific studies of lacustrine carbonates. While Lake Van setting and chemistry are very different from the lakes you are working with, these papers highlight the suboptimal suitability of bulk carbonate samples for paleoenvironmental reconstruction and elucidate which factors can compromise the bulk signal.

The works by McCormack, Kwiecien and co-authors are certainly relevant when discussing the environmental signal in bulk carbonate and merit to be cited here. The relevance of carbonate mineralogy has been added in this context, referring to the suggested studies (Lines 40-44):

"Changes in the bulk carbonate composition can be caused by variations in the assemblage of calcifying organisms, in clastic carbonate input, or in the amount of inorganic calcite precipitated in the lake (e.g. Bright et al. 2006, Hammarlund & Buchardt 1996). Furthermore, carbonate minerals such as calcite, aragonite or dolomite differ in their isotopic composition, so a potentially variable carbonate mineralogy both related to coeval sedimentation and to early diagenesis must be taken into account (McCormack & Kwiecien 2021, McCormack et al. 2018)."

I really like that the conclusions loop back to the relevant goals listed in the introduction. Having said that I find the conclusion misleadingly presented. I agree that differences in vital offset -corrected d18O values of different carbonate components suggest different periods of formation and might point to the amplitude of seasonal temperature contrasts. This holds true only if several components are extracted from the same sedimentary layer and their isotopic composition is compared and contrasted (conclusion 1). However, this information is interwoven with influences of lake water d18O and temperature. By the time the reader reaches conclusion 2, the essential notion of comparison is already forgotten, and it reads like any seasonal change in water temperature is clearly reflected in d18O of any biogenic carbonate, and I cannot agree with this statement. The order of arguments provided in conclusion 2 does not strengthen it either. Please, streamline the arguments towards the conclusion, not away from it. Again, a well-designed sketch in the introduction, could help in making this conclusion more succinct. Conclusion 3, while correct, is very loosely formulated and, in

its present form reiterates the findings of McCormack & Kwiecien 2021. Your work deals with a more complex example and is the first such comprehensive attempt of comparing carbonate components from shallow water, above the thermocline of an open lake (as explained in conclusion 2). I think that focusing conclusions on this particular case and making them more specific will be very beneficial.

We thank the reviewer for these helpful comments to focus the conclusions and to be clearer about the specific conditions under which these conclusions are valid. We have revised the conclusions with these suggestions in mind.

In conclusion 1 (Line 496) we have added a reference to the newly added sketch in introduction (Figure 1).

In conclusion 2, we repeat the essential notion of comparison between species as the basis for seasonal water temperature reconstruction (Line 497-499):

"...it is possible to estimate seasonal water temperature changes from the d18O of lake water and of specific biogenic carbonates, given that different components from the same sediment layer which were formed during different seasons are analyzed separately. [...] seasonal changes in lake water temperature are clearly reflected in the d18O of multiple biogenic carbonates."

Furthermore, we made a clearer distinction between our conclusion 2 and the "but", or the argument away from our conclusion, as Reviewer 1 calls it, which must be considered when studying longer time scales (Line 500-506):

"Under such conditions, seasonal changes in lake water temperature are clearly reflected in the d18O of biogenic carbonates through their effect on isotope fractionation between water and carbonate. Water temperature changes could still exert the dominant control on carbonate d18O on longer (decadal to millennial) time scales, where significant changes in air temperature can be expected. However, additional factors must be considered in that case, such as the effects of changes in d18O of atmospheric precipitation, brought about by related or independent climate dynamics, or potential variations in evaporative enrichment in the d18O of the lake water (changing evaporation/inflow ratio of the lake)."

Lastly, we focus conclusion 3 on the specific case of our study site (Line 508-512):

"The intra-specific variability in d18O and d13C of biogenic carbonates highlights that care must be taken to obtain representative subsamples of a species for each time interval, especially when environmental conditions such as water temperature can change rapidly in shallow water. The

inter-specific variability observed in Lake Locknesjön adds to the growing body of evidence demonstrating the importance of performing measurements on specific carbonate components of lake sediments rather than on bulk carbonate."

**Specific comments:**

**Abstract**

Line 4: 'lake water and water temperature'

Corrected.

Lines 21-25: this info is correct but as a 'textbook knowledge' is unsuitable for the abstract

This information has been removed from the abstract and included in the introduction instead (see reply to the first general comment and new Figure 1).

**Introduction**

Line 36: 'depending on the local context'

Corrected.

Line 40: remains of lacustrine organisms

Corrected.

Lines 40-45: open lakes are more prone to calcite than aragonite precipitation, but carbonate mineralogy also plays a role in bulk carbonate d18O composition. Please, check McCormack et al., 2018

The reference has been added (see reply to the general comment above).

Line 76: their - whose?

The sentence has been rewritten to "The causes of vital effects […]"

**Material and methods**

Line 177-178: were the valves visually checked for organic matter remains? Was the potential organic matter left intact?

The valves were visually checked for remains of organic matter or other contaminations. The organic matter was not kept for analyses. We have rephrased the sentence to (Lines 184-185):

> "A number of living specimens of ostracods and Pisidium were
> found in the sediment samples. These were briefly boiled so that
> organic matter could be removed easily from the valves."

Line 184-185: valves? I was under impression that gastropods have shells and operculum but not valves

Pisidium is a bivalve, not a gastropod, and their shell consists of two valves. We have added this information the first time Pisidium is mentioned in the introduction (Line 103):

> "the following types of samples were taken [...]: the aragonitic
> shells of the bivalve mollusk Pisidium sp."

**Results**

Lines 235-239: this is interpretation, not result

It is true that at a few instances the Results section is not purely descriptive but presents our results along with an interpretation. However, we prefer to keep the present structure, as this Results section presents the results on water isotopes and carbonate isotopes in individual species (with some explanations). The following Discussion section then focuses on the comparison of different species in the context of a potential paleo-temperature interpretation of these proxies, including an understanding of the climate signal in the lake water d18O. To make this approach clear, we have added an introductory paragraph to the Results section (Lines 224-227):

> "This Section presents the results of our stable isotope
> measurements of water and biogenic carbonates, and explains the
> influences on their isotopic composition. The subsequent
> Discussion then evaluates in how far these influences can be
> constrained through our "snapshot" approach by comparing
> different species, and which paleoclimate information can
> potentially be obtained from the carbonates."

Line 268: 'surface sediment' is misleading if it refers only to encrustations collected from the surface but not to the bulk surface sediment

The expression has been changed to (Lines 282):

> "subfossil encrustations collected from surface sediments".

Lines 267-275: information provided here is correct, but it is not a result

See comment on Lines 235-239.

Line 286: and what about autochthonous carbonates? Can you exclude/ discuss their presence?

Indeed, we should compare the d13C values of fine calcite to both (inorganically precipitated) autochthonous carbonates and allochthonous carbonates. The d13C

values indicate that the fine calcite is composed of disaggregated chara encrustations (mean d13C of -1.48 permil), which are influenced by the fractionation due to photosynthesis. The d13C of the bedrock in the catchment, potential source of allochthonous carbonate, is higher (mean of -0.5 permil; see Line 399). The d13C of other autochthonous carbonates, which are not influenced by photosynthesis, is lower (around -4 permil or lower for ostracods and Pisidium; not measured for inorganic). We have modified the sentence as follows (Lines 299-303):

```
"The observed d13C values are similar for fine calcite and
encrustations. This indicates that the fine calcite, which also
makes up the matrix of the carbonate-rich sediments of Lake
Locknesjön, is composed of disaggregated Chara hispida
encrustations. Allochthonous carbonates would be expected to have
a higher d13C than the fine calcite, close to the d13C of bedrock
in the catchment (around -0.5 permil, see Section 4.3). Other
autochthonous carbonates which are not influenced by
fractionation due to photosynthesis have a lower d13C."
```

Lines 308-313: information provided here is correct, but it is not a result

See comment on Lines 235-239.

Lines 326-329: information provided here is correct, but it is not a result

See comment on Lines 235-239.

Line 326: exobiotic mentioned for the first time without explanation

It is true that the term "exobenthic" is not common vocabulary, and is actually not necessary here to explain that the d13C of DIC controls the d13C of ostracod valves. We have therefore taken it out.

**Discussion**

Lines 344-346: correct information but should be better explained in the introduction (see general comments)

This sentence has been deleted here and the influencing factors on carbonate d18O are now explained in more detail in the introduction, including the new Figure 1 (see reply to general comments above).

Lines 449-453: without clear reference to a figure, I cannot see how your results demonstrate these two points. Also, the points are very vague - what do you mean by 'sufficiently large'? How do you know or how can you test what is an 'representative average'? Please, try to rethink this argument.

We now included a reference to Figure 10 to make clear what the variability in d18O could mean in terms of a temperature reconstruction. It is true that the terms "sufficiently

large" and "representative average" are vague. On the other hand, the number of samples needed to be representative depends on the variability between individual samples. We reformulated the sentences as follows (Lines 463-468):

"However, our results demonstrate that the range of reconstructed temperatures from such individual samples is large (Figure 10), whereas the temperature reconstructed from an average of all samples per species/instar is in good agreement with observations. Consequently, it must be assured that isotopic records obtained from lacustrine biogenic carbonates for the study of past climate changes yield representative values for the time span covered by each analyzed sediment interval, through either (i) sufficiently large numbers of individual samples, or (ii) homogenized samples incorporating sufficiently large numbers of specimens."

**Figures**

All figures are informative but with small adjustment they could convey the message more efficiently.

Fig. 2: Please, indicate clearly 8 m water depth mark (the deepest sampling point). If the grid is necessary in the figures, please, align the legend within the grid boxes. Also, please put the data points in the foreground not in the background. The present effect is visually unsettling.

(Figure 3 in the revised manuscript) A shading has been added in Fig. 3a to indicate the water depth range where carbonate samples were taken for this study. In both panels, the data points have been moved to the foreground, the legend has been aligned with the grid and the axis range set to integers.

Fig. 3: Please, align the legend within the grid boxes. I am not sure if the symbols in the upper left and right corner of figure 3a are intended?

(Figure 4 in the revised manuscript) The legends have been aligned in the top left corner of each panel. The symbols were intended and are now explained in the caption.

Fig. 4: Please, make the data points in panel 4a larger, they are barely visible. Similarly, the triangles in panel 4b

(Figure 5 in the revised manuscript) The size of the data points has been increased in panels 5a and 5b.

Fig. 5: Please, unify the scales in fig a and b (panel b is visibly horizontally stretched, although the range of the values is the same) also the ticks on the d13C axis are suboptimally distributed, if taking the grid into consideration (with the grid values at -10, -8.75, -7.5 and so on).

(Figure 6 in the revised manuscript) The scales in the figure have been adjusted and the tick marks set at integers.

Fig. 6: Please unify the scales in fig a and b (panel b is visibly horizontally stretched although the range of the values is the same). The legend is a bit confusing; it took me a while to figure out what am I looking at. 'Sediment sample' even if explained in the legend is misleading, why not calling it 'dead fragments' or 'subfossil fragments'?

(Figure 7 in the revised manuscript) The scales have been adjusted accordingly. The sample type "sediment sample" has been re-named to `subfossil fragment` and the entire manuscript has been checked for consistency with this terminology.

Fig. 7: The grid is distracting. If the authors want to keep the grid why not stopping at full intervals (e.g.: -7.5, -2.5 for d13C and -8, -4 for d18O) rather than cutting it of randomly?

(Figure 8 in the revised manuscript) The axis ranges have been adjusted for a cleaner grid layout.

Fig. 8: What are exactly 'dead' and 'living' samples? Are the fragments of encrustation described as 'sediment sample' in the legend of figure 6 considered 'dead'? Please, define the term and use it consistently.

(Figure 9 in the revised manuscript) In agreement with the comment on Fig. 6, the sample types have been re-named in this figure to `living` and `subfossil fragments`. These terms are now defined in the Methods Section (2.3), and the entire text, figures and captions were revised to keep a consistent terminology.

Fig. 9: The same comment as above about the grid

(Figure 10 in the revised manuscript) The grid layout has been adjusted.

Table 1: The species, instar and the no. samples are the same for both panels, I suggest merging them into one. For the consistency, I would suggest adding all data presented in figure 7 (including 'fine calcite', 'fragmented encrustation from surface sediments' and Chara samples from Lake Blaktjärnen). Please, also indicate if these are measured or vital offset -corrected data. Last comment here - please try to keep the terminology consistent throughout the main text, figures and figure captions and the table.

We have changed the arrangement of the table putting the values for d18O and d13C for each sample type on one line. A sentence has been added in the table caption to indicate that the numbers are the measured values, not corrected for vital offsets. To keep the table legible, we prefer not to give separate values for living and subfossil samples for all species. The given values are also the ones which are referred to and discussed in the text. All data, i.e. measurements of all individual samples, will be provided as a data set in Pangaea and linked to this article. Lastly, we have revised the manuscript to keep a consistent terminology regarding "living" and "dead" samples, see reply to comment on Fig. 8.

To wrap up, I think this is a really valuable contribution showing pitfalls of using single carbonate component and highlighting the interpretational difficulties but, also benefits of multi-component analyses, and I very much wish to see it published. I hope that authors will find my feedback helpful.

Best wishes, Ola Kwiecien

---

## Author Comment (AC2)

**A modern snapshot of the isotopic composition of lacustrine biogenic carbonates – Records of seasonal water temperature variability**

Inga Labuhn, Franziska Tell, Ulrich von Grafenstein, Dan Hammarlund, Henning Kuhnert, and Bénédicte Minster

**RC2**

The study by Labuhn et al. discusses an interesting issue of applicability of oxygen stable isotope measurements in specific lacustrine carbonates in reconstructions of past water temperatures. The study adds to the already existing knowledge as pointed out by authors.

I found the manuscript well written and interesting. The introduction is informative and points out the key information based on the available literature sources. The data are well presented with high-quality graphics. The authors discuss the possible mechanisms that control the stable isotope composition of the carbonates studied and explain the possible reasons for the differences in the stable isotope composition of encrustations and shells.

The study confirms the established knowledge that due to the differences in stable isotope composition $\delta^{18}O$ measurements should be performed on the specific types of carbonates instead of bulk carbonate samples of unknown and potentially time-variable composition. The most important outcome of the study is showing that by studying selected carbonates it is possible to estimate seasonal water temperature changes.

Despite the overall good quality of the study I suggest considering the specific comments listed below before the manuscript can be accepted for publication.

We thank Reviewer 2 for her evaluation of the manuscript and the constructive comments. Our replies and modifications in the manuscript are explained after each comment (in blue).

**Specific comments:**

Line 4: change 'on' to 'by'

Corrected.

Line 98: delete double 'the'

Corrected.

Lines 145-148: I would not limit the growth of Chara to May-July. What about August and possibly also September? You suggest that charophytes studied are perennial.

It is true that the charophytes can be perennial. The sentence was misleading, because it referred to the meteorological conditions in the year of sampling. The samples were taken in July, i.e. the August and September temperatures of that specific year do not have any influence. Since the reference to Chara growth is not necessary in this paragraph about meteorology, we have deleted the sentence.

Lines 156-157: What was the bottom area (cm2) where each of the surface sediment was sampled?

We thank the reviewer for pointing out the missing details on the sampling procedures. We have included the size of the bottom area (8x8 cm2) in Line 156.

160-162: Information about the sampling of charophytes is lacking. Were the whole macroalgae taken? Cut at the water-surface sediment interface? How many individuals of each species were sampled?

These details on charophyte sampling have been added in Lines 158-159: the whole algae were sampled, up to six individuals which were cut off by hand by the divers.

Lines 168-169: Since I was involved in the studies of the isotopic composition of recent charophytes I have also tried to remove organics with H2O2. I have never managed to remove all. Part of the stem was always resilient and remained after the treatment.

This is an important issue. It is crucial that the carbonate is not contaminated by organic matter, as carbonate and organic matter differ in their isotopic composition. We cut off branchlets and internodes from the charophytes prior to the H2O2 treatment (see Line 162-163). These small parts were free from organic matter after the treatment. All samples were inspected under a binocular prior to isotope measurements. Examples of clean encrustations from our living charophyte samples after H2O2 treatment are shown in the images below. The inside of the encrustation, where the stem used to be, is clearly free from any organic matter.

[Figure]

[Figure]

Line 169: Please explain what the 'fine-grained calcite sticking to the encrustations' is. Why did you remove it? How do you know that you did not remove a fraction of encrustations at the same time?

We do not know the origin of this fine calcite. The material might originate from the encrustations themselves, or it could be inorganically precipitated calcite deposited on the encrustations. For this reason, we measured its isotopic composition separately from the encrustations, in order to evaluate whether it could represent a contamination and bias the results of the isotope measurements on encrustations. Our results indicate that the fine calcite actually is composed of disaggregated Chara encrustations, because the d13C values are very similar (see Lines 282-285).

To explain the reason for these measurements, we have added the following information (Lines 176-177):

"The removed fine calcite was also kept for isotope measurements in order to test if it is composed of inorganically precipitated calcite, or if it originates from the encrustations themselves."

Lines 255-257: I guess both living specimens and Pisidium shells taken from surface sediments in fact originate from the surface sediments therefore it is better to say: shells of living Pisidium specimens and empty Pisidium shells

Indeed, our previous terminology was imprecise, because both the "dead"/empty Pisidium shells as well as the living specimens are found in the sediment. In agreement with the comment of Reviewer 1 on Fig. 6, we have changed this paragraph to (Lines 268-270):

"There is no significant difference in isotope values between the valves of living specimens sampled in 2018 and the subfossil valves (i.e. which were formed and deposited during previous years), possibly because their time spans of carbonate

accumulation overlap. The mean d18O is the same, and the mean
d13C differs by 0.19 permil."

Furthermore, we have revised the entire manuscript, figures and captions to keep a consistent terminology regarding the different types of samples.

Line 257: probably it would be good to change 'living and dead samples' to 'shells of living and dead mussels'

We agree with the reviewer and changed the terminology accordingly; see comment above.

Lines 263-264 and 380-382: Charophytes – you studied 'single stalks from an internode or branchlet'. Encrustations at one specimen are not formed at the same time but as charophytes grow. Therefore in the isotopic studies of charophytes, specific fragments were studied, e.g. apical fragments. The variation of stable isotope values of charophyte encrustations studied may result from the fact that different fragments had CaCO3 precipitated at slightly different times, i.e. as the charophyte grew. In my opinion, the larger isotopic range of Chara hispida is also due to the gradual seasonal growth and precipitation of encrustations in the changing ambient conditions.

We thank Reviewer 2 for this comment and the following ones related to Chara growth and the timing of calcification. These aspects are important to consider in the interpretation of the isotopic composition of the encrustations, but did not become sufficiently clear in the previous version of the manuscript. A number of changes have been made accordingly (see also the following two comments).

Lines 279-281: "Each analyzed sample consisted of only one internode
or branchlet, which means that environmental variations within
the growing season or between different years are not averaged in
individual samples. The samples represent only the short period
of time when the respective part of the Chara grew."

Lines 392-395: "Chara can be perennial, but the samples measured in
this study, encrustations from single internodes or branchlets,
were precipitated at different times as the Chara grew, i.e.
individual samples may represent different times within a growing
season and possibly different years, hence explaining their
larger isotopic range."

Lines 264-265: Do you have confirmation that Chara hispida from the lake studied overwintered? Charophytes are not always perennial. Overwintering can occur but it is not a rule. I have observed this during the field studies I participated in. You can have a look at publications e.g. of Mariusz PeÅ echaty – an experienced charophyte scientist with extensive field experience, in which the issue is discussed.

We do not have any confirmation from our field observations that the studied Chara overwintered. We therefore have added the information that Chara can be perennial, but that this is not always the case, and added the suggested reference (Lines 276-277):

"Chara can be annual or perennial (Martin et al. 2003, Pełechaty et al. 2013). It is not known whether overwintering occurs in Locknesjön, but different parts of the algae are formed during different times of the growing season, or possibly during different years."

Lines 265-266: Whole new and several dozen cm high charophyte can grow within one season –personal field observations.

It is true that charophyte growth can be fast. The idea of this sentence was to say that the calcification occurs only around the part of the Chara which is growing, i.e. the apices. To be clearer, also considering the changes in the above comment related to annual/perennial growth, we have modified the sentence to (Lines 277-278):

"The growth of Chara is apical, i.e. calcification at a given point in time occurs only around these apices (Coletta et al. 2001)."

Line 266: Which internodes and branchlets were sampled? Apical ones or fragments from different parts of charophytes. Also, what were the sizes of charophytes? How tall were the macroalgae studied? This information is important in the context of the discussion. Thick and dense charophyte stands can form a specific microhabitat, they can also limit the extent of water mixing to the bottom.

We sampled fragments from different parts distributed throughout the charophytes. They were about 20 cm tall. The density of charophyte stands at the sampling locations in Locknesjön was relatively low, and we suppose that it does not have an effect on the isotopic composition of the encrustations. We do not observe any significant trend in d18O or d13C with depth (see Fig. 9), and the charophytes became very sparse at 7 m depth and absent at 8 m depth.

We have added information on the sampled fragments in the Material and Methods section:

Lines 173-174: "Internode and branchlet samples were cut off from different parts of the living Chara [...]."

Lines 158-159: "The density of Chara hispida stands at the sampling locations was relatively low. The Chara population became very sparse at 7 m depth and absent at 8 m depth."

Lines 279-282: This difference may result from the intensity of photosynthesis and density of charophyte patches.

We have added the intensity of photosynthesis (associated with fast calcite precipitation) as a reason for these offsets (Lines 295-297):

"Other studies, on the contrary, have found significant offsets from equilibrium due to kinetic effects during intense

```
photosynthesis    associated    with    fast    calcite    precipitation
(Apolinarska et al., 2016; Andrews et al., 2004)."
```

See comment above about the density of charophyte stands.

Lines 283-284: Here 'fine calcite' is mentioned once again, what kind of calcite is that? More explanation is needed.

This indeed needs an explanation, which we have added in Lines 176-177 (see our reply to the comment on Line 169).

Lines 317-319: Influence of stratification is mentioned here however, previously in the manuscript it was stated what waters within the epilimnion are well mixed and looking at the data one can see that thermocline is below the deepest site sampled. Also, see lines 345-348 and most important lines 352-357: These fragments confirm my concern about interpreting $\delta_{18}O$ values in Candona as related to water stratification.

We thank the Reviewer for pointing out this lack of clarity. We do suppose that the water in the epilimnion is well mixed, and that all our sampling locations down to 8 m water depth are above the summer thermocline. These assumptions are supported by the water temperature profiles (Fig. 3) and the isotopic composition of the water (Fig. 5, Fig. 9). Therefore, stratification is not an explanation for the differences in d18O between ostracod instars A-4/-3/-2 and instars A-1. They all form their valves during a time of stratification (in the epilimnion, above the thermocline), and the seasonally changing temperature within the epilimnion causes differences in d18O.

On the other hand, stratification - or the absence thereof - becomes important for adult ostracods, as they form when isotopically enriched surface water is mixed with deep water with a lower d18O (although the lower water temperature at this time is the dominant influence on the valve carbonate d18O). We have re-written this paragraph as follows (Lines 331-338):

```
"Juvenile ostracods form their valves when the lake is thermally
stratified during the warm season. As the instars A-2 to A-4 form
at the time of maximum water temperature, the mean d18O of their
valves is significantly lower than in A-1 and adult samples.
There  is  no  significant  difference  between  A-2,  A-3  and  A-4
instars.  The  average  d18O  of  A-1  samples  is  higher  than  in
younger   instars   because   A-1   instars   develop   when   water
temperature starts to decrease in autumn. Moreover, by the end of
the summer the water in the epilimnion is more enriched in 18O by
evaporation compared to the time when the younger instars calcify
their  valves,  leading  to  a  further  increase  in  the  d18O  of  A-1
valves.  The  valves  of  adult  ostracods  start  forming  in  late
autumn.  When  overturning  occurs  in  the  lake,  stratification
ceases  and  the  isotopically  enriched  surface  water  is  mixed  with
deep water with a lower d18O."
```

Lines 324-329: Also, $\delta_{13}C$ of adult ostracods is lowest because of water mixing and return of the 13C-depleted DIC of the waters from below the thermocline

This explanation has been added to the paragraph (Lines 345-347):

"When adult valves are formed, the lake is not stratified and the overturning brings DIC with a lower d13C from depths below the summer thermocline to the surface."

Lines 424-425: Which is an apparent drawback.

We agree with Reviewer 2, because the quantification of the temperature influence on carbonate d18O would require continuous measurements of water temperature throughout the period of carbonate precipitation in a sample. A part of the motivation for our study was to test this "snapshot approach" and evaluate whether relevant information to aid the interpretation of paleoclimate records from sediment cores can be obtained without long-term monitoring. Even if continuous water temperature measurements were available, it may be difficult to link the precise timing of calcification in biogenic carbonates to a specific water temperature, because the calcification does not occur continuously throughout the lifetime of the organism.

437-438: Temperatures absolutely unlikely to occur. In central Europe, even during days with temperatures > 30 during the day, water temperature in the epilimnion reaches 24-25oC.

We agree that these temperatures certainly do not occur in lake water in our study area. The sentence has been re-written replacing "unlikely" with "impossible" (Line 451).

Fig. 3a: Why don't you present the complete, i.e. whole year precipitation and temperature data for the year of sampling – 2018? This may differ from the long term data. In fact, the difference is already visible, especially in precipitation values.

We have added the monthly temperature and precipitation data of the year 2018 from January to July (which corresponds to the time of sampling in 2018), in Figure 3a.

Sincerely,

Karina Apolinarska

---

## Author Response (AR2)

**Authors' Response**

We thank the Editor and Reviewer for their comments on the revised manuscript. The authors' response and explanations of changes in the manuscript are written in blue after each comment.

Comments to the author:
Dear authors,
Thank you for revising your manuscript.
I sent it to one of the reviewers for further evaluation, and the feedback is good. Please see below their minor revision requests.

I also went through your revisions, and have few points to raise that I hope you could address prior to accepting your paper for publications.

Overall, I would suggest few words/compounds replacement for conciseness and for consistencies:
1. Replace "atmospheric precipitation: with "rainfall", this way the use of the earlier confusing "precipitation" is solved

It is true that the term "atmospheric precipitation" is long, but since both "atmospheric precipitation" and "carbonate precipitation" are often repeated in the text, it is important to specify in each context which precipitation is meant (see earlier comment by Reviewer 1). "Rainfall" could be a shorter alternative, but a significant part of the atmospheric precipitation falls as snow in our study area, which is why we prefer to keep "atmospheric precipitation".

2. The word "account for" is a phrasal verb and makes the manuscript wordy. It is used several times in the manuscript. I encourage the authors to replace as much of it with "consider"

We replaced "taken into account" by "considered".

Few detailed comments are below.

Best wishes,
Dr. Ny Riavo Voarintsoa
BG Associate Editor

Figure 1
It is confusing to read increase in precipitation in that figure "an increase in air temperature leads to an increase in precipitation d18O" --> what precipitation, I believe one the reviewer has pointed this out already, but it was not addressed carefully. Also, I

wanted to know is this air temperature–precipitation relationship always straightforward, any relevant reference that support that statement?

"Precipitation" has been changed to "atmospheric precipitation" for consistency with the rest of the text. The relationship between temperature and precipitation d18O is dependent on the climatological context, but they are clearly correlated in mid- and high latitudes. This information was included in the figure caption with an appropriate reference.

I think the linkage between isotopic composition of the water and air temperature is a bit blurry, and I highly recommend the author to consider the degree of evaporation and the relative humidity, rather than simply temperature.

It is already mentioned in the figure under point (1) that higher temperatures are associated with stronger evaporation from the lake. We added relative humidity as an influencing factor in the figure caption.

The author caution readers to note that there is an opposite influence of temperature on carbonate d18O in (1) and (2); but this is not what I understand from looking at that figure. In (1) it only shows lake water (there is no carbonates), and in (2) it indicates O isotope fractionation between the two phases. I believe the authors should be clearer, or rewrite the figure caption to avoid such misinterpretation from the readers?

It is true that (1) and (2) point to lake water and water-carbonate fractionation, respectively, and not to carbonates directly. The effect on the carbonates is a consequence of the effects on (1) lake water and on (2) fractionation. To be clearer, we rephrased the figure caption.

L: 68: when you mention "isotopic equilibrium", I'd like to caution you in using that expression (see Daëron, M., Drysdale, R.N., Peral, M. et al. Most Earth-surface calcites precipitate out of isotopic equilibrium. Nat Commun 10, 429 (2019). https://doi.org/10.1038/s41467-019-08336-5)

We added the suggested reference and a sentence to explain that isotopic equilibrium rarely occurs in Earth surface carbonates.

L. 165: Rephrase: a diving team collected some water samples, surface sediments, and living Chara from the lake.

The sentence was re-written accordingly.

Reviewer comments:

Dear authors,

Thank you for revising your manuscript and addressing my and other reviewer's comments with care and detail. You have delivered an elegant work and important scientific contribution.
My last and very minor comment: two sentences in conclusions are correct but read awkwardly, below I suggest how to rephrase them:

Both sentences were rephrased according to the reviewer's suggestion.

Line 522 – 524: Based on the understanding of these environmental controls, it is possible to estimate seasonal water temperature changes from the d18O of lake water and of specific biogenic carbonates, provided that components formed during different seasons but sampled from the same sediment layer are analysed individually

Line 535 – 537: The intra-specific variability in d18O and d13C of biogenic carbonates highlights that care must be taken to obtain representative subsamples of a species for each time interval, especially if working with shallow water environments where water temperature can change rapidly over short time.